# End-to-End Co-Optimization of Adaptive $k$-space Sampling and Reconstruction for Dynamic MRI

**George Yiasemis** [1,2] **Jan-Jakob Sonke** [1,2] **Jonas Teuwen** [1,2,3]

G.YIASEMIS, J.SONKE, J.TEUWEN@NKI.NL

[1] *Netherlands Cancer Institute* [2] *University of Amsterdam* [3] *Radboud University Medical Center*

**Editors:** Accepted for publication at MIDL 2026

## Abstract

Accelerating dynamic MRI is essential for advancing clinical imaging and improving patient comfort. Most deep learning methods for dynamic MRI reconstruction rely on predetermined or random subsampling patterns that are uniformly applied across all temporal frames. Such strategies ignore temporal correlations and fail to optimize sampling for individual cases. To address this, we propose E2E-ADS-Recon, an end-to-end framework for adaptive dynamic MRI subsampling and reconstruction. The framework integrates an Adaptive Dynamic Sampler (ADS), which generates case-specific sampling patterns for a given acceleration factor, with a dynamic MRI reconstruction network that reconstructs the adaptively sampled data into a dynamic image sequence. The ADS can produce either frame-specific or unified patterns across time frames. We evaluate the method on multi-coil cardiac cine MRI data under both 1D and 2D sampling settings and compare it with standard and optimized non-adaptive baselines. E2E-ADS-Recon achieves superior reconstruction quality, particularly at higher acceleration rates. These results highlight the benefit of case-specific adaptive sampling and demonstrate the potential of joint sampling–reconstruction optimization for dynamic MRI. Code and trained models will be made publicly available upon acceptance.

**Keywords:** Adaptive MRI Sampling, Dynamic MRI Reconstruction, Joint Optimization.

## 1. Introduction

Magnetic Resonance Imaging (MRI) represents an important clinical imaging modality. However, its inherently slow acquisition and susceptibility to motion-stemming from respiratory, bowel, or cardiac movements-pose significant challenges, especially for dynamic imaging. These factors hinder the collection of fully-sampled dynamic frequency-domain data, known as $k$-space, leading to degraded image quality. Consequently, patients are often required to perform actions like voluntary breath-holding to minimize motion effects. To address these issues, accelerated dynamic acquisition techniques, which involve subsampling of the $k$-space below the Nyquist ratio (Shannon, 1949), have been employed.

Recent advancements have seen Deep Learning (DL)-based MRI reconstruction methods significantly outperforming traditional approaches (Lustig et al., 2008; Gamper et al., 2008). These DL methods excel in reconstructing MRI images from highly-accelerated scans (Guo et al., 2023; Yiasemis et al., 2022). Despite the predominant focus on static MRI reconstruction due to limited reference dynamic acquisitions, recent developments have extended to dynamic MRI reconstruction (Lønning et al., 2024; Qin et al., 2019; Yiasemis et al., 2024b; Yoo et al., 2021; Zhang et al., 2023).

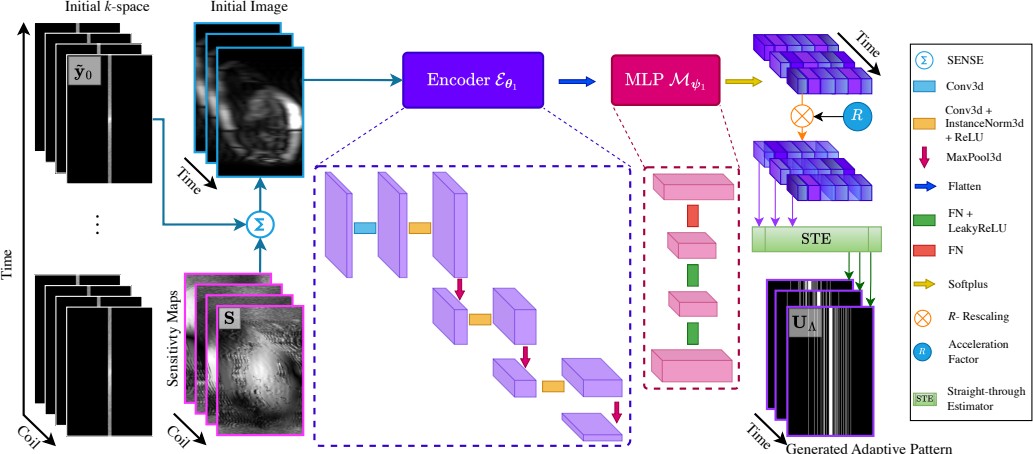

Figure 1: Overview of the Adaptive Dynamic Sampler for frame-specific $k$-space sampling. Initial multi-coil dynamic $k$-space data $\tilde{\mathbf{y}}_0$ and sensitivity maps $\mathbf{S}$ are used to perform a SENSE reconstruction, which serves as the input to the encoder $\mathcal{E}_{\boldsymbol{\theta}_1}$ in the ADS module. The encoder output is flattened and passed to a multi-layer perceptron $\mathcal{M}_{\boldsymbol{\psi}_1}$, which generates probability vectors for each potential sampling location. These probabilities are softplused, rescaled to meet the acceleration factor $R$, and binarized using a straight-through estimator layer to produce the sampling pattern $\mathbf{U}_\Lambda$, ensuring $\mathtt{AF}(\Lambda) = R$. For brevity here we assume that $\tilde{\mathbf{y}}_0$ comprises only of ACS data and we also assume only a single ADS cascade ($N = 1$).

Nonetheless, a critical limitation persists in DL-based dynamic MRI reconstruction: current methodologies typically rely on predetermined, often arbitrary, subsampling patterns, uniformly applied across all temporal frames. This approach overlooks the benefits of exploiting temporal correlations present in adjacent frames and the potential superiority of DL-based learned adaptive sampling schemes, which constitutes our primary motivation.

Previous studies have demonstrated the feasibility of learning optimized on training data sampling schemes at specific acceleration factors, trained jointly with a reconstruction network in single (Bahadir et al., 2019) or multi-coil (Zhang et al., 2020) settings. In the context of adaptive sampling, various DL-based sampling approaches have been proposed in both single (Pineda et al., 2020; Van Gorp et al., 2021; Yin et al., 2021; Bakker et al., 2020) and multi-coil (Bakker et al., 2022; Gautam et al., 2024) scenarios.

Among the existing literature, most approaches remain limited to static imaging. One study (Shor, 2023) extended this direction to dynamic acquisitions by learning a single, non-adaptive optimized non-Cartesian trajectory jointly with a reconstruction model. To our knowledge, this remains the only example explicitly targeting dynamic sampling from training data. Yet, it does not explore *adaptiveness*, nor does it support varying acceleration rates, leaving the field of adaptive dynamic Cartesian subsampling unexplored. Our work aims to fill this gap. Our contributions can be summarized as follows:

- We introduce E2E-ADS-Recon, a novel end-to-end *learned adaptive dynamic subsampling and reconstruction method* for dynamic, multi-coil, Cartesian MRI data. This method integrates an adaptive dynamic sampling model with a sensitivity map prediction module and a dynamic MRI reconstruction method, all trained *end-to-end*. Our approach is designed for simultaneous training across *varying acceleration factors*.

- We propose two dynamic sampling methodologies: one that learns distinct adaptive trajectories for each frame (*frame-specific*) and another that learns a uniform adaptive trajectory for all frames (*unified*), with both methods applied to 1D and 2D sampling.
- We evaluate our pipeline on a multi-coil, dynamic cardiac MRI dataset (Wang et al., 2023). Our evaluations include comparisons with pipelines where dynamic subsampling schemes are dataset-optimized or use fixed or random schemes.

## 2. Background and Methods

### 2.1. Accelerated Dynamic MRI Acquisition and Reconstruction

Given a series of fully-sampled dynamic multi-coil $k$-space data $\mathbf{y} \in \mathbb{C}^{n \times n_c \times n_f}$, the underlying dynamic image $\mathbf{x}^* \in \mathbb{C}^{n \times n_f}$, can be obtained by applying the inverse Fast Fourier transform $\mathcal{F}^{-1}$: $\mathbf{x}^* = \mathcal{F}^{-1}(\mathbf{y})$, where $n = n_1 \times n_2$, $n_c$, and $n_f$ represent the spatial dimensions, the number of coils, and the number of frames (time-steps) of the dynamic acquisition, respectively. To accelerate the acquisition, the $k$-space is subsampled. The acquired subsampled dynamic multi-coil $k$-space $\tilde{\mathbf{y}} \in \mathbb{C}^{n \times n_c \times n_f}$ can be defined by the forward problem for $k = 1, .., n_c$, $\ t = 1, .., n_f$:

$$\tilde{\mathbf{y}}_t^k = \mathcal{A}_{\Lambda^t, \mathbf{S}^k}(\mathbf{x}_t^*) + \tilde{\mathbf{e}}_t^k, \quad \mathcal{A}_{\Lambda^t, \mathbf{S}^k}^k := \mathbf{U}_{\Lambda^t} \mathcal{F} \mathbf{S}^k, \tag{1}$$

where $\mathbf{U}_M$ denotes a subsampling operator acting on a set $M \subseteq \Omega$ as follows:

$$\mathbf{z}_M : (\mathbf{z}_M)_i = (\mathbf{U}_M \mathbf{z})_i = \mathbf{z}_i \cdot \mathbb{1}_M(i), \quad i \in \Omega := \{1, 2, \cdots\}, \quad \mathbf{z} \in \mathbb{C}^{n \times n_c}. \tag{2}$$

Here, $\Omega$ denotes the sample space comprising all possible sampling options, where $|\Omega|$ equals $n_2$ or $n$ for 1D (column) sampling and 2D (point) sampling, respectively. $\mathcal{F}$ denotes the forward FFT, $\mathbf{S}^k \in \mathbb{C}^{n \times n}$ the sensitivity map of the $k$-th coil, and $\mathbf{e}_t^k$ measurement noise. The acceleration factor (AF) of the acquisition of $\tilde{\mathbf{y}}$ is determined by the dynamic acquisition set $\Lambda = \{\Lambda^t\}_{t=1}^{n_f} \subset \Omega^{n_f}$: $\text{AF}(\Lambda) = \frac{n_f |\Omega|}{\sum_{t=1}^{n_f} |\Lambda^t|}$, where $|\cdot|$ denotes the cardinality.

The goal of dynamic MRI reconstruction involves obtaining an approximation of $\mathbf{x}^*$ using $\tilde{\mathbf{y}}$, formulated as a regularized least-squares optimization problem (Bertero, 2006):

$$\hat{\mathbf{x}} = \underset{\mathbf{x} \in \mathbb{C}^{n \times n_f}}{\arg\min} \frac{1}{2} \sum_{t=1}^{n_f} \sum_{k=1}^{n_c} \left\| \mathcal{A}_{\Lambda^t, \mathbf{S}^k}^k (\mathbf{x}_t) - \tilde{\mathbf{y}}_t^k \right\|_2^2 + \mathcal{G}(\mathbf{x}) \tag{3}$$

where $\mathcal{G} : \mathbb{C}^{n \times n_f} \to \mathbb{R}$ represents an arbitrary regularization functional imposing prior reconstruction information.

Our objectives in this project involve learning an adaptive dynamic sampling pattern $\mathbf{U}_\Lambda := (\mathbf{U}_{\Lambda^1}, \cdots, \mathbf{U}_{\Lambda^{n_f}})$ that maximizes the information content of the acquired subsampled data $\tilde{\mathbf{y}}$ which is adapted based on some initial dynamic $k$-space data $\tilde{\mathbf{y}}_{\Lambda_0}$, and within the same framework train a DL-based dynamic reconstruction technique. The idea is that both sampling and reconstruction are improved and co-trained by exploiting cross-frame information found across the dynamic data.

### 2.2. Sensitivity Map Prediction

Coil sensitivities are estimated using the fully-sampled central $k$-space region, or autocalibration signal (ACS). This is further refined by a 2D U-Net-based (Ronneberger et al., 2015)

Sensitivity Map Predictor (SMP), $\mathcal{S}_{\boldsymbol{\omega}}$, an approach proven effective in enhancing sensitivity map prediction (Sriram et al., 2020; Peng et al., 2022; Yiasemis et al., 2022).

## 2.3. Adaptive Dynamic Sampler

To adapt the subsampling pattern to each case, we introduce the Adaptive Dynamic Sampler (ADS), extending a previous approach for static adaptive sampling (Bakker et al., 2022) trained jointly with a variational network reconstruction model (Sriram et al., 2020). Given initial measurements $\tilde{\mathbf{y}}_0 \in \mathbb{C}^{n \times n_c \times n_f}$ acquired from an initial set $\Lambda_0 \subset \Omega^{n_f}$ (e.g., $\Lambda_0 = \Lambda_{\mathrm{acs}}$), ADS allocates a sampling budget online: $\mathbf{n_b} = (n_b^1, \cdots, n_b^{n_f}) = (\frac{n_a}{R} - |\Lambda_0^1|, \cdots, \frac{n_a}{R} - |\Lambda_0^{n_f}|)$. Here, $n_a = |\Omega|$ denotes the total number of potential samples, and $R$ an arbitrary acceleration factor. We focus on frame-specific sampling (for unified set $n_f = 1$).

ADS operates through a number of $N$ cascades, dividing the sampling budget evenly as $\frac{\mathbf{n_b}}{N}$ across cascades. Each cascade comprises an encoder module $\mathcal{E}_{\boldsymbol{\theta_m}}$ and a multi-layer perceptron (MLPs) $\mathcal{M}_{\boldsymbol{\psi_m}}$. The encoders follow a U-Net (Ronneberger et al., 2015) encoder structure, alternating between $l_{\mathrm{enc}}$ 3D convolutional layers ($3^3$ kernels) with instance normalization(Ulyanov et al., 2017) and ReLU (Xu, 2015) activations, and max-pooling layers ($2^3$ kernels), except for the first layer. MLPs consist of $l_{\mathrm{mlp}}$ linear layers, with leaky ReLU (Xu, 2015) (with negative slope 0.01) activation, except for the final layer.

Each $\mathcal{E}_{\boldsymbol{\theta_m}}$ receives (subsampled) multi-coil $k$-space measurements as input, which are reconstructed into a single image via complex conjugate sensitivity map sum (SENSE reconstruction). The resulting image is subsequently flattened and introduced into $\mathcal{M}_{\boldsymbol{\psi_m}}$, generating probability vectors $\mathbf{p}_m = (\mathbf{p}_m^1, \cdots, \mathbf{p}_m^{n_f}) \in \mathbb{R}^{n_f \times n_a}$ such that $\mathbf{p}_m^t \in \mathbb{R}^{n_a}$. Probabilities in $\mathbf{p}_m^t$ corresponding to previously acquired measurements on $\bigcup_{j=0}^{m-1} \Lambda_j^t$ are set to zero. A softplus activation function is applied to each $\mathbf{p}_m^t$, followed by rescaling (see Algorithm 1) to ensure $\mathbb{E}(\mathbf{p}_m^t) = \frac{n_b^t}{N \times n_a}$.

To enable differentiability of the binarization process and allow end-to-end training, we apply a straight-through estimator (Yin et al., 2019) (STE) for gradient approximation, following prior work (Bakker et al., 2022; Yin et al., 2021; Zhang et al., 2020). This stochastically generates $\Lambda_m^t$ by binarizing $\mathbf{p}_m^t$ through rejection sampling to meet the sampling budget $\frac{\mathbf{n_b}}{N}$, with gradients approximated using a sigmoid function (slope = 10). The STE's forward and backward passes are detailed in Algorithms 2 and 3, with further details in Appendix B.1.1.

The first ADS cascade processes the initial data $\tilde{\mathbf{y}}_0$, and each subsequent cascade $m$ takes as input $k$-space data $\tilde{\mathbf{y}}_{m-1}$ produced from the previous cascade aiming to produce a new acquisition set $\Lambda_m \subset \Omega^{n_f}$. The new $k$-space data is then acquired from the predicted set $\Lambda_m$ and is subsequently aggregated with previous data, equivalent to an acquisition on $\bigcup_{j=0}^m \Lambda_j$. This sequential approach yields the final set $\Lambda := \bigcup_{m=0}^N \Lambda_m \subset \Omega^{n_f}$ satisfying $\mathtt{AF}(\Lambda) = R$. See Fig. 1 for a depiction of the ADS module, with further details in Algorithms 4 (frame-specific sampling) and 5 (unified sampling).

## 2.4. Dynamic MRI Reconstruction

Our proposed pipeline (Section 2.5) is designed to be independent of the underlying reconstruction algorithm. Throughout this work, we primarily employ vSHARP (Yiasemis et al., 2025, 2024b), a state-of-the-art (SOTA) model that achieved leading performance in the

CMRxRecon Challenges (Lyu et al., 2025; Wang et al., 2025). vSHARP efficiently solves (3) via half-quadratic variable splitting and ADMM-unrolled optimization over $T$ iterations.

To verify that our pipeline generalizes beyond a single reconstruction backbone, we additionally integrate MEDL-Net (Qiao et al., 2023), another SOTA method that addresses the same variational problem through an iterative architecture with dense inter-cascade connections, enabling fewer cascades without loss of fidelity.

Given subsampled measurements $\tilde{\mathbf{y}}$ and sensitivity maps $\mathbf{S}$, the reconstruction model predicts the underlying dynamic image: $\hat{\mathbf{x}} = \mathcal{R}_\phi(\tilde{\mathbf{y}}; \mathbf{S})$. Further algorithmic details can be found in the related literature (Yiasemis et al., 2025, 2024b; Qiao et al., 2023).

### 2.5. End-to-end Adaptive Dynamic Sampling and Reconstruction

For our end-to-end approach we integrate the methodologies detailed in Sections 2.2, 2.3, and 2.4. The process is visually summarized in Fig. 2 and algorithmically in Algorithm 6. Given ACS data $\tilde{\mathbf{y}}_{\mathrm{acs}}$, the sensitivities $\mathbf{S}$ are predicted via $\mathcal{S}_{\boldsymbol{\omega}}$. Subsequently, with initial measurements $\tilde{\mathbf{y}}_0$ and a specified acceleration factor $R$, the adaptive dynamic sampling module $\mathrm{ADS}_{\boldsymbol{\psi}, \boldsymbol{\theta}}$ generates an adaptive dynamic subsampling pattern $\mathbf{U}_\Lambda$, where $\Lambda$ is as defined in Section 2.3. Note that for acceleration consistency we choose $\Lambda_{\mathrm{acs}} \subseteq \Lambda_0$. After acquiring the data $\tilde{\mathbf{y}}_\Lambda$ using $\mathbf{U}_\Lambda$, $\mathcal{R}_\phi$ reconstructs a dynamic image utilizing $\mathbf{S}$ and $\tilde{\mathbf{y}}_\Lambda$.

The training of our framework proceeds in an end-to-end manner, ensuring gradients propagate effectively across all three modules, allowing for simultaneous optimization of sampling and reconstruction. The ADS module, as detailed in Section 2.2, employs a STE to approximate gradients through the binarization process, ensuring differentiability and enabling reconstruction loss gradients to flow back through the ADS module. The co-optimization of ADS with SMP and the reconstruction model is achieved through loss computation at the end of the forward pass of the pipeline. In this design, the reconstruction network processes adaptively sampled measurements, and the resulting reconstruction loss (see Section 3.3 for loss calculation) propagates supervision signals throughout the pipeline. This joint optimization ensures that both the sampling patterns and reconstruction quality are systematically improved. Implemented in PyTorch (Paszke et al., 2019), the framework leverages its automatic differentiation capabilities to compute gradients seamlessly across all components, including the probabilistic sampling steps within ADS.

## 3. Experiments

### 3.1. Dataset

We utilized the CMRxRecon dataset (Wang et al., 2023), which comprises 473 scans from distinct patients, consisting of 4D multi-coil cine cardiac $k$-space data (a total of 3,185 2D dynamic slices). The data were split into training, validation, and test sets (251, 111, 111 scans, respectively). For more details refer to Appendix B.2.1. In addition, we further evaluated generalization on an unseen ventricular outflow tract / aortic cine dataset from CMRxRecon 2025 (Xu, 2025), which differs in anatomical view and motion characteristics and was not observed during training (44 scans).

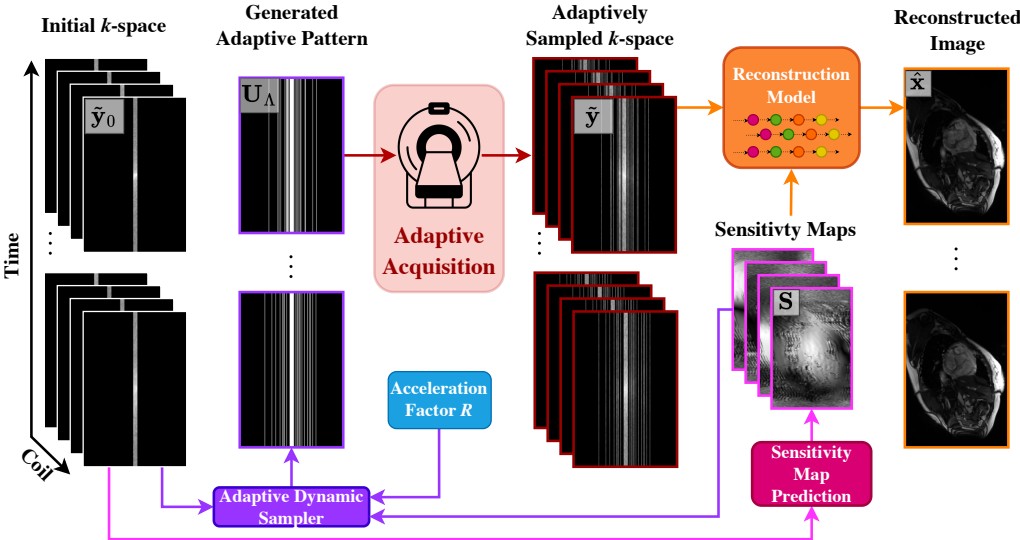

Figure 2: Overview of the E2E-ADS-Recon pipeline for frame-specific $k$-space sampling. Initial multi-coil dynamic $k$-space data $\tilde{\mathbf{y}}_0$ include ACS data $\tilde{\mathbf{y}}_{\text{ACS}}$, which are used by the SMP module to generate coil sensitivity maps $\mathbf{S}$. These sensitivity maps and the initial measurements are input to the ADS module, which generates adaptive sampling patterns $\mathbf{U}_\Lambda$ based on the desired acceleration factor $R$. These patterns guide subsampled $k$-space acquisitions during dynamic imaging. The subsampled data $\tilde{\mathbf{y}}$ are processed with the sensitivity maps $\mathbf{S}$ in the reconstruction module, yielding reconstructions $\hat{\mathbf{x}}$. The pipeline, including ADS, SMP and reconstruction model, is jointly optimized end-to-end to enhance imaging quality. For simplicity, the illustration assumes a single ADS cascade ($N = 1$).

### 3.2. Comparative & Ablation Studies

To validate our E2E-ADS-Recon, particularly with respect to adaptive sampling, we evaluate various sampling strategies under both frame-specific and unified settings:

**Learned**:

(a) Our pipeline employing different initializations: (i) ACS-initiated ($\Lambda_0 = \Lambda_{\text{acs}}$) (Adpt-AcsInit), (ii) equispaced with $\frac{|\Omega|}{|\Lambda_0|} = R - 4$ (Adpt-EqInit-I), (iii) $\frac{|\Omega|}{|\Lambda_0|} = R - 2$ (Adpt-EqInit-II), and (iv) $kt$-equispaced with $\frac{|\Omega|}{|\Lambda_0|} = R - 2$ (Adpt-$kt$EqInit-II). For configurations (ii)-(iv), $\Lambda_{\text{acs}} \subset \Lambda_0$, and these are applicable only to 1D sampling.

(b) Non-adaptive optimized learned schemes, where the sampling space is parameterized and optimized end-to-end with the reconstruction model (Bahadir et al., 2019) (Opt).

**Predetermined/Random**:

(a) Common non-adaptive schemes, including (i) 1D random uniform (Rand), (ii) 1D Gaussian (Gauss-1D), (iii) 2D equispaced (Equi), (iv) 2D Gaussian (Gauss-2D), and (v) radial (Rad) trajectories (Yiasemis et al., 2024c). In frame-specific experiments, a unique trajectory was used for each frame, whereas in unified settings, the same pattern was applied across all frames.

(b) Non-adaptive schemes with temporal interleaving ($kt$ schemes) (Tsao et al., 2003), including $kt$-equispaced ($kt$Equi), $kt$-Gaussian 1D ($kt$Gauss-1D) and $kt$-radial ($kt$Rad).

We provide more information in Appendix B.2.3

For non-adaptive methods, we replace the ADS with each sampling strategy while keeping the rest of the architecture (SMP and reconstruction model) unchanged.

We also conduct ablation studies on the E2E-ADS-Recon model with the following choices:

1. A modified version of our proposed method using a single cascade for ADS ($N = 1$) instead of two (comparative studies), also employing different initializations.
2. Non-uniform sampling budget allocation across time frames in the ADS module (Adpt-NU), compared to equal division used in the original framework (applicable only in frame-specific settings).

### 3.3. Experimental Setup

**Optimization** Models were developed in PyTorch (Paszke et al., 2019), using Adam (Kingma, 2014) with a learning rate starting at 1e-3, linearly increasing to 3e-3 over 2k iterations, then reducing by 20% every 10k iterations, over 52k iterations. Experiments were conducted on single A6000 or A100 NVIDIA GPUs, with a batch size of 1. We used a dual-domain loss strategy (Yiasemis et al., 2025), combining image and frequency domain losses:

$$\mathcal{L} = \sum_{j=1}^{T} w_j \bigg[ \sum_{t=1}^{n_f} \big( \mathcal{L}_{\text{SSIM}}(\hat{\mathbf{x}}_t^{(j)}, \mathbf{x}_t^*) + \mathcal{L}_1(\hat{\mathbf{x}}_t^{(j)}, \mathbf{x}_t^*) + \mathcal{L}_{\text{HFEN}}(\hat{\mathbf{x}}_t^{(j)}, \mathbf{x}_t^*) \big)$$
$$+ \mathcal{L}_{\text{SSIM3D}}(\hat{\mathbf{x}}^{(j)}, \mathbf{x}^*) \bigg] + 3 \cdot \mathcal{L}_{\text{NMAE}}(\hat{\mathbf{y}}, \mathbf{y}), \quad w_j = 10^{(j-T)/(T-1)} \tag{4}$$

where $\{\hat{\mathbf{x}}^{(j)}\}_{j=1}^T$ denotes the predicted dynamic images from vSHARP's unrolled steps, and $\hat{\mathbf{y}}$ represents the predicted $k$-space data. The choice of individual loss components and their respective weights follows the training protocol of the vSHARP applied to the CMRxRecon challenge 2023 (Yiasemis et al., 2025, 2024b; Lyu et al., 2025). The definitions of each loss component can be found in Appendix B.2.4.

**Hyperparameter Settings** In our adaptive sampling experiments, we configured the ADS sampler with $N = 2$ cascades. Unless specified otherwise, we use image domain encoding ADS modules. Our setup features encoders with $l_{\text{enc}} = 3$ scales and MLPs with $l_{\text{mlp}} = 3$ layers each. In all our experiments the SMP module comprised a 2D U-Net with 4 scales (16, 32, 64, 128 channels) and we used vSHARPs (Yiasemis et al., 2024b) with $T = 8$ and 3D U-Nets as denoisers composed of 4 scales (16, 32, 64, 128 channels).

**Reconstruction Model Robustness** We repeat the (frame-specific) comparative studies outlined in Section 3.2 using MEDL-Net (Qiao et al., 2023) as the reconstruction model instead of vSHARP to explore the robustness of our end-to-end pipeline. Optimization and hyperparameter choice details are specified in Appendix B.3.

**Subsampling** All experiments (learned or otherwise) used a fraction $r_{\text{acs}} := |\Lambda_{\text{acs}}| = 4\%$ of $\Omega$ to fully sample the $k$-space center, denoted as $\tilde{\mathbf{y}}_{\Lambda_{\text{acs}}}$, which is used for sensitivity map prediction, similar to the literature (Sriram et al., 2020; Peng et al., 2022; Yiasemis et al., 2022). In learned sampling experiments, $\Lambda_0 = \Lambda_{\text{acs}}$, unless stated otherwise. During training the acceleration was randomly chosen between $4\times, 6\times, 8\times$, while for inference we evaluated each setup on acceleration factors of $4\times, 6\times$ and $8\times$. In addition, we evaluated all trained models at higher acceleration factors ($10\times$ and $12\times$) not seen during training to assess extrapolation behavior.

**Evaluation** The models were assessed using SSIM, PSNR, and NMSE metrics, as defined in the literature (Yiasemis et al., 2024c). These metrics were averaged per slice or frame within each scan, after being centrally cropped (region of interest, ignore background) to two-thirds of each dimension. For significance testing, we use the almost stochastic order test (Dror et al., 2019; Ulmer et al., 2022) with $\alpha = 0.05$ (see Appendix B.2.6).

## 4. Results

Quantitative analysis is presented in Figures 3, S1 and S2, which display the distributions of SSIM, PSNR, and NMSE metrics for the test set. Boxplots highlight the top-performing methods and their statistical significance across all configurations.

Our results consistently demonstrate that frame-specific sampling outperforms unified sampling across all tested configurations, acceleration factors, and sampling dimensions (line-1D, point-2D). Consistent trends were also observed on an unseen aortic cine dataset with different anatomy and motion patterns as illustrated in Figure S4.

Among the methods evaluated, our proposed E2E-ADS-Recon achieves the highest performance across nearly all combinations of frame-specific and unified setups, for both 1D and 2D sampling. E2E-ADS-Recon shows statistically significant results in most metrics, with the highest SSIM values achieved by variants of our approach. A similar trend is observed for PSNR and NMSE, although in frame-specific 1D sampling at $R = 4$, equispaced sampling slightly outperforms our method. When evaluated at higher, unseen acceleration factors ($R = 10, 12$), the same qualitative performance ordering was preserved, with adaptive sampling maintaining its advantage over non-adaptive baselines (Figure S3).

For 1D sampling, both frame-specific and unified setups benefit from combining our adaptive strategy with equispaced initialization and ACS $k$-space data, yielding the highest average performance. Despite this, non-adaptive equispaced sampling remains the top competitor. At high acceleration factors ($6\times, 8\times$), all ADS configurations generally outperform non-adaptive methods in both frame-specific and unified settings concerning the SSIM metric, as well as the PSNR and NMSE in most cases.

In 2D setups, E2E-ADS-Recon significantly surpasses non-adaptive methods across all metrics and acceleration factors, with optimized parameterized sampling being the closest competitor.

Qualitative results are shown in Figure 4, which shows examples of generated 1D sampling patterns for both setups. Visual inspection of generated sampling patterns reveals that learned patterns tend to prioritize lower-frequency components near the $k$-space center, while occasionally incorporating higher frequencies. In Figure 5, we illustrate example reconstructions from the unified experiments at a high acceleration factor ($R = 8$), where adaptive methods demonstrate improved preservation of anatomical structures and reduced artifacts relative to non-adaptive approaches. Additional pattern examples and image reconstructions for all setups are available in Appendix D.

Inference runtime is reported in Table S1. Across all configurations, mean inference times remain within a narrow range of approximately 10.5–11.5 seconds per test volume. Adaptive methods introduced a small additional overhead (less than 1 second on average) relative to non-adaptive baselines, primarily due to the sampling prediction step. Uni-

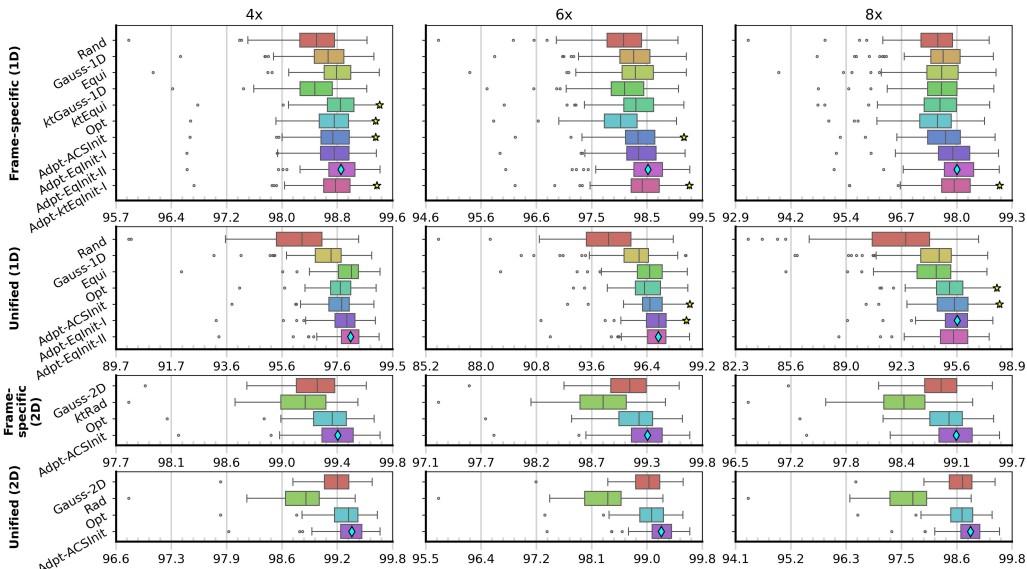

Figure 3: SSIM (×100) metrics across all experimental settings and setups. Diamonds (◇) on the box-plot median indicate the average best methods. A star (⋆) on the upper whisker indicates non-significance in comparison to the average best method.

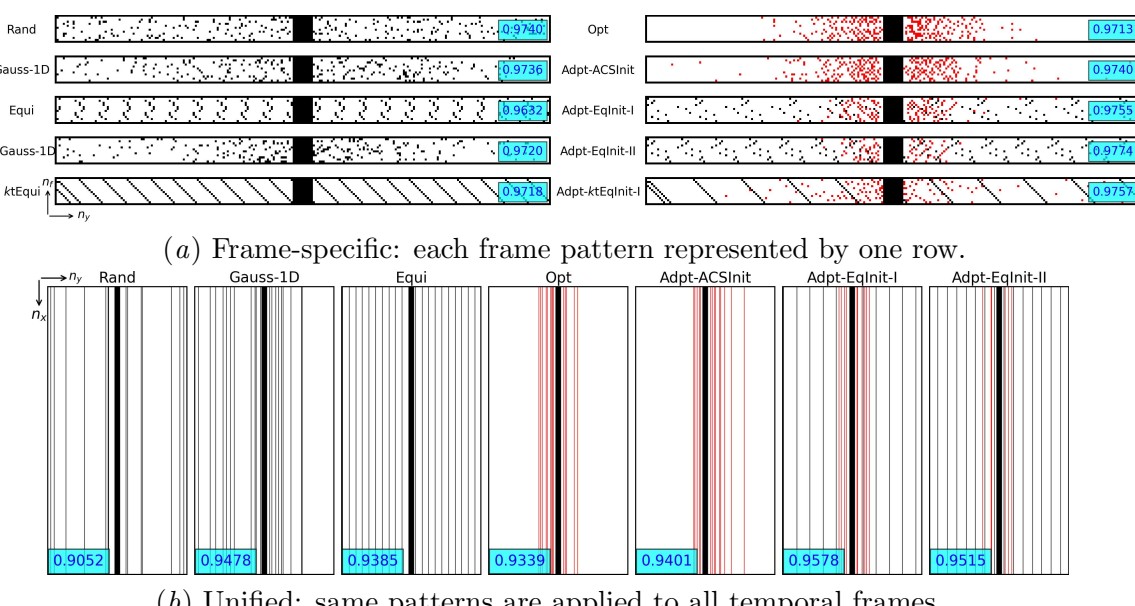

(*a*) Frame-specific: each frame pattern represented by one row.

(*b*) Unified: same patterns are applied to all temporal frames.

Figure 4: Examples of 1D patterns across setups at $R = 8$ for (a) frame-specific and (b) unified settings. Black: fixed/initial, red: learned pattern. Cyan boxes mark SSIM values.

fied compared to frame-specific adaptive sampling setups exhibit faster runtime overall, as expected given their shared sampling mask across frames.

**Reconstruction Model Robustness** Results are provided in Appendix C.4, where we replace the reconstruction model with MEDL-Net. Figures S5, S6 and S7 present the corre-

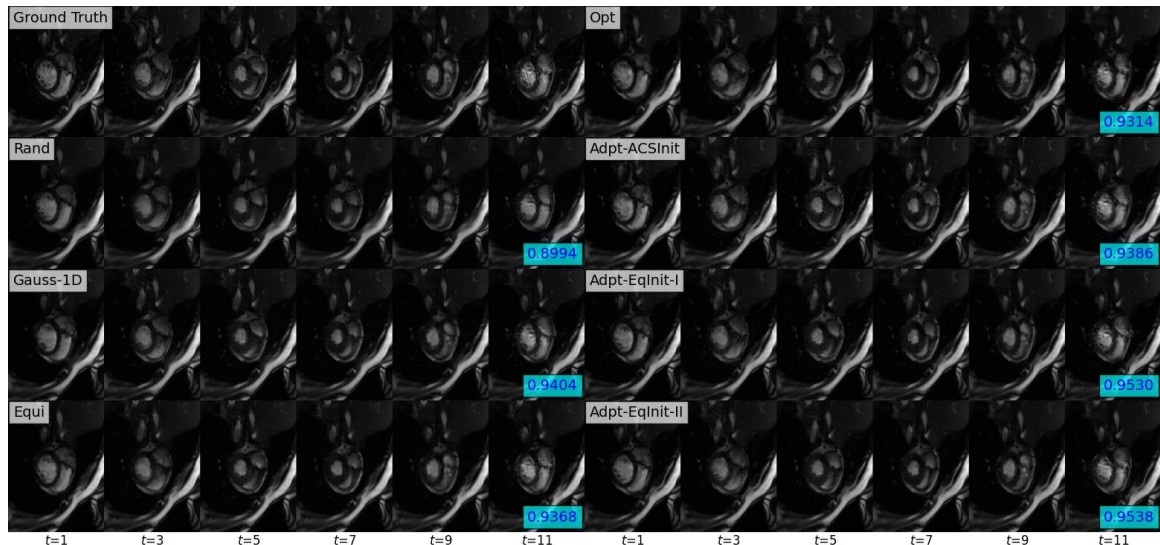

Figure 5: Example of reconstructions across setups for unified 1D sampling at $R = 8$.

sponding SSIM, PSNR, and NMSE metrics. Across both 1D and 2D sampling, the numerical trends remain consistent with those observed using vSHARP as the reconstruction module. In nearly all acceleration-metric combinations, adaptive sampling methods outperform their non-adaptive counterparts on average, except for SSIM at $R = 4$, where the $k$tEqui scheme shows a slight, though statistically insignificant, advantage. For 1D sampling, E2E-ADS-Recon with equispaced initialization continues to be the top-performing configuration in most cases. Furthermore, the benefits of adaptive sampling become more evident at higher acceleration factors (e.g., $R = 8$). Additionally, while not the primary focus of this study, we observe that employing vSHARP yields overall better results than MEDL-Net.

**Ablation Studies** Tables S2 and S3 provide results from our ablation study employing single ($N = 1$) ADS cascade, as well as non-uniform frame-specific adaptive sampling. Using a single cascades yields performance comparable to using two cascades, with most cases outperforming non-adaptive methods. Non-uniform sampling budget allocation did not show competitive results compared to equal budget distribution across time frames.

## 5. Conclusion and Discussion

In this work, we introduced E2E-ADS-Recon, integrating a deep learning-based adaptive sampler for dynamic MRI accelerated acquisition with state-of-the-art reconstruction networks for dynamic reconstruction. E2E-ADS-Recon is trained end-to-end to generate adaptive, case-specific dynamic acquisition patterns for any given acceleration factor, followed by dynamic image reconstruction, with the aim of exploiting temporal correlations between temporal frames.

We evaluated E2E-ADS-Recon using 4D fully-sampled cine cardiac MRI data under two sampling settings: frame-specific (distinct subsampling patterns per time frame to exploit inter-frame correlations) and unified (the same pattern applied across all frames). We also considered two sampling dimensions: 1D (line) and 2D (point). Our approach was benchmarked against equivalent pipelines that replaced the adaptive sampler of E2E-

ADS-Recon with other various non-adaptive sampling schemes, including common 1D or 2D schemes (random, equispaced, Gaussian, radial), as well as data-optimized approaches.

Visualization of the sampling patterns (see Figure 4 and Appendix D) generated by E2E-ADS-Recon revealed that the ADS component consistently produced adaptive patterns with denser sampling in the lower frequency regions (closer to the $k$-space center) which contain a structural information and image contrast information, while occasionally sampling higher frequencies. In addition, the visualizations show that in frame-specific sampling experiments, the ADS module learned to generate distinct patterns for each temporal frame. As a result, this allowed the reconstruction model to effectively leverage complementary information across frames and produce better reconstructions in comparison to the unified setting.

Even though the performance improvements observed with E2E-ADS-Recon may appear incremental, they are statistically significant. Reporting such improvements is a common practice in MRI literature, where the focus often lies on incremental yet meaningful enhancements (e.g., close performance of top teams in CMRxRecon Challenge 2023 (Lyu et al., 2025)). The relatively modest gains in reconstruction quality are partially attributed to the powerful vSHARP model used, which inherently minimizes the impact of different sampling patterns. Nonetheless, the consistent performance gains across experiments highlight the potential of adaptive sampling, particularly at higher accelerations.

Additionally, we replaced the vSHARP model with another state-of-the-art method (MEDL-Net) to assess the reconstruction model's robustness. The results remained consistent, with adaptive sampling configurations continuing to outperform non-adaptive approaches across sampling dimensions and acceleration factors.

Despite the promising results of E2E-ADS-Recon, its supervised training requires fully-sampled data, similar to other data-optimized or adaptive sampling approaches, limiting its applicability to the availability of such data. Non-learned sampling methods, on the other hand, can leverage subsampled datasets for self-supervised learning of reconstruction models without requiring fully-sampled data. Recent studies have shown that joint supervised and self-supervised training of reconstruction models can enhance self-supervised results (Yiasemis et al., 2024a). Future work could explore such direction to extend the applicability of E2E-ADS-Recon to datasets without fully-sampled $k$-spaces.

In addition, since all data were acquired at a single site using the same scanner and protocol, the model was evaluated under in-distribution conditions. As a result, its robustness to out-of-distribution variations-such as differences in scanner vendors, acquisition protocols, or patient populations-remains untested. Future work should assess generalization under more heterogeneous, clinically representative conditions.

Another factor to consider is the use of initial data (e.g., ACS) in E2E-ADS-Recon. The ADS module relies on this initialization to generate adapted, case-specific acquisition, unlike optimized sampling methods that do not require pre-sampled data. This dependency on the initial sampling pattern is a limitation shared with methods requiring initial data for calibration, such as GRAPPA or RAKI, where the quantity and size of the initial data significantly affect downstream reconstruction (Knoll et al., 2020). Similarly, in learned adaptive sampling in static imaging scenarios, initialization with low-frequency (ACS) data is common, as seen in approaches employing reinforcement learning (Pineda et al., 2020; Bakker et al., 2020) or deep learning (Yin et al., 2021). Additionally, the influence of

different sampling patterns on reconstruction performance has been well-documented in prior studies (Yiasemis et al., 2024c; Hammernik et al., 2017). In E2E-ADS-Recon, using ACS-reconstructed images for initialization is a natural choice, as these data are already required for sensitivity profile estimation in DL-based reconstruction methods (Yiasemis et al., 2022).

While our retrospective study demonstrates the conceptual potential of E2E-ADS-Recon, practical implementation within an MRI scanner remains untested and is beyond the scope of this work. Implementing such an end-to-end framework on a scanner would require substantial engineering and integration efforts; however, this study serves as an initial step toward investigating how sampling and reconstruction might be co-optimized. Importantly, the observed improvements, though small and evaluated retrospectively, suggest that adaptive sampling may offer meaningful gains if effectively translated into prospective clinical settings.

Although the runtime overhead introduced by adaptive sampling was modest (see Table S1), real-time deployment would impose additional constraints. In particular, frame-specific adaptive methods require generating a new sampling pattern for each frame–slice combination within a limited temporal window. The dataset used in this study was acquired with a temporal resolution of 50 milliseconds per cardiac frame (see Appendix B.2.1). As reported in Section 4, the estimated upper bound on per-frame overhead is approximately 27 milliseconds, suggesting that our current implementation is compatible with this timescale under retrospective conditions. However, actual deployment would introduce additional sources of latency, including scanner–module communication and real-time synchronization, which remain to be addressed in future work.

Our results underscore the advantages of frame-specific sampling over unified strategies. Frame-specific methods, which employ distinct patterns per time frame, more effectively leveraged temporal correlations, resulting in higher reconstruction quality. However, practical challenges such as large gradient switches and eddy currents might arise with frame-specific schemes, whereas unified approaches are less likely to encounter these issues. Additionally, the implementation of frame-specific sampling requires knowledge of the anatomical phase at each moment to apply the correct sampling pattern. Finally, while our results indicate that 2D sampling outperforms 1D sampling consistently with previous studies (Yin et al., 2021; Yiasemis et al., 2024c), future studies should investigate the physical implementation and application of 2D sampling in a clinical setting.

Overall, while our study illustrates the potential benefits of adaptive sampling in deep learning–based dynamic MRI reconstruction, future work is needed to evaluate its feasibility in prospective, real-time acquisition settings.

## Acknowledgments

This work was supported by institutional grants of the Dutch Cancer Society and of the Dutch Ministry of Health, Welfare and Sport. The authors acknowledge the Research High Performance Computing (RHPC) facility of the Netherlands Cancer Institute (NKI) for the computational resources.

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

## End-to-End Co-Optimization of Adaptive $k$-space Sampling and Reconstruction for Dynamic MRI – Supplementary Material

## Appendix A. Algorithms

### A.1. Rescale

---

**Algorithm 1** Rescale to budget

---
**Input**   : Predicted probabilities $\mathbf{p} \in \mathbb{R}^{n_a}$, Sampling budget `budget`
**Output:** Rescaled probabilities $\mathbf{p}_{\text{rescaled}}$
$s \leftarrow \text{budget}/n_a$ ;                            // Calculate target sparsity based on budget
$\bar{p} \leftarrow \mathbb{E}(\mathbf{p})$ ;                                    // Compute mean probability
$r \leftarrow s/\bar{p}$ ;                                    // Scaling factor for each sample
$\beta \leftarrow (1-s)/(1-\bar{p})$ ;                        // Adjustment factor for non-scaled elements
**if** $r \leq 1$ **then**
$\quad\mid\quad$ $\mathbf{p}_{\text{rescaled}} \leftarrow \mathbf{p} \cdot r$ ;                              // Scale down to fit target sparsity
**end**
**else**
$\quad\mid\quad$ $\mathbf{p}_{\text{rescaled}} \leftarrow \mathbf{1} - (\mathbf{1} - \mathbf{p}) \times \beta$ ;          // Adjust larger values while preserving
$\quad\mid\quad$ distribution
**end**
**return** $\mathbf{p}_{\text{rescaled}}$ such that $\mathbb{E}(\mathbf{p}_{\text{rescaled}}) = s$

---

## A.2. Straight-through Estimator

---

**Algorithm 2** Forward Pass of Straight-through Estimator

---

**Input**  : Rescaled Probability $\mathbf{p} = (p^1, \cdots, p^{n_f}) \in \mathbb{R}^{n_f \times n_a}$, Tolerance $\texttt{tol} = 1e-3$
**Output:** Binarized pattern $\mathbf{r}$
**for** $t \leftarrow 1 \; to \; n_f$ **do**

    **for** *each* $p_i^t$ *in* $p^t$ **do**

        Initialize $\pi_i \sim U(0,1)$ ;        `// Generate random uniform probability for` $\pi_i$
        $r_i^t \leftarrow \mathbb{I}(p_i^t > \pi_i)$ ;    `// Binarize` $p_i^t$`: Set` $r_i^t = 1$ `if` $p_i^t > \pi_i$`, otherwise` $r_i^t = 0$

    **end**

    **while** $|\mathbb{E}[r^t] - \mathbb{E}[p^t]| \geq tol$ **do**

        **for** *each* $p_i^t$ *in* $p^t$ **do**

            $\pi_i \sim U(0,1)$ ;        `// Regenerate random uniform probability for` $\pi_i$
            $r_i^t \leftarrow \mathbb{I}(p_i^t > \pi_i)$ ;        `// Re-binarize` $p_i^t$ `with the new` $\pi_i$

        **end**

    **end**

**end**
$\mathbf{r} = (r^1, \cdots, r^t)$
 **store** $\boldsymbol{\pi}$ ;                        `// Store` $\boldsymbol{\pi}$ `for backward pass`
**return r**
 `// Note: For unified sampling set` $n_f = 1$

---

---

**Algorithm 3** Backward Pass of Straight-through Estimator

---

**Input**  : Gradient of loss w.r.t. output $\nabla \mathcal{L}$, Input probability $\mathbf{p}$, Probability $\boldsymbol{\pi}$ from forward
        pass, Slope $\texttt{slope} = 10$,
**Output:** Gradient w.r.t. input vector $\nabla \mathbf{p}$
Compute the sigmoid gradient:

$$\mathbf{g} \leftarrow \frac{\texttt{slope} \cdot \exp(-\texttt{slope} \cdot (\mathbf{p} - \boldsymbol{\pi}))}{(1 + \exp(-\texttt{slope} \cdot (\mathbf{p} - \boldsymbol{\pi})))^2}$$

$\nabla \mathbf{p} \leftarrow \mathbf{g} \cdot \nabla \mathcal{L}$ ;    `// Compute the gradient with respect to input probability p`
**return** $\nabla \mathbf{p}$
 `// Note: All operations are element-wise`

---

## A.3. Adaptive Dynamic Sampler

---

**Algorithm 4** Adaptive Dynamic Sampling for Frame-specific Patterns

---

**Input** : Initial data $\tilde{\mathbf{y}}_0 \in \mathbb{C}^{n \times n_c \times n_f}$, Acceleration $R$, Sample Space $\Omega$
**Output:** Acquired $k$-space data $\tilde{\mathbf{y}}$ for $R$
$n_a \leftarrow |\Omega|$;
$\mathbf{n_b} = (n_b^1, \cdots, n_b^{n_f}) \leftarrow (\frac{n_a}{R} - |\Lambda_0^1|, \cdots, \frac{n_a}{R} - |\Lambda_0^{n_f}|)$ ;                // Calculate total budget
**for** $m \leftarrow 1$ *to* $N$ **do**
$\quad$ $\mathbf{p}_m \leftarrow \mathcal{M}_{\boldsymbol{\psi}_m} \circ \mathcal{E}_{\boldsymbol{\theta}_m}(\tilde{\mathbf{y}}_{m-1})$ ;         // Produce adaptive probabilities $\mathbf{p}_m \in \mathbb{R}^{n_f \times n_a}$
$\quad$ **for** $t \leftarrow 1$ *to* $n_f$ **do**
$\quad\quad$ **for** *each* $i \in \left( \bigcup_{j=0}^{m-1} \Lambda_j^t \right)$ **do**
$\quad\quad\quad$ $(\mathbf{p}_m^t)_i \leftarrow 0$ ;                              // Zero-out already sampled indices
$\quad\quad$ **end**
$\quad\quad$ $\mathbf{p}_m^t \leftarrow \text{Softplus}(\mathbf{p}_m^t)$ ;                                    // Apply Softplus
$\quad\quad$ $\mathbf{p}_m^t \leftarrow \text{Rescale}(\mathbf{p}_m^t; \texttt{budget} = n_b^t/N)$ ;        // Rescale such that $\mathbb{E}(\mathbf{p}_m^t) = \frac{n_b^t}{N \times n_a}$
$\quad\quad$ $\Lambda_m^t \leftarrow \text{STE}(\mathbf{p}_m^t)$ ;                              // Produce adaptive sampling
$\quad$ **end**
$\quad$ $\Lambda_m \leftarrow (\Lambda_m^1, \cdots, \Lambda_m^{n_f})$;
$\quad$ $\mathbf{z}_m \leftarrow \mathbf{U}_{\Lambda_m}\mathbf{y}$ ;                              // Acquire new $k$-space on $\Lambda_m$
$\quad$ $\tilde{\mathbf{y}}_m \leftarrow \tilde{\mathbf{y}}_{m-1} + \mathbf{z}_m = \mathbf{U}_{\cup_{j=0}^m \Lambda_j}\mathbf{y}$ ;        // Aggregate with previous data $\tilde{\mathbf{y}}_{m-1}$
**end**
$\tilde{\mathbf{y}} \leftarrow \tilde{\mathbf{y}}_N$;
**return** $\tilde{\mathbf{y}}$
// Note:  Lines 6-7 ensure $\Lambda_{m-1}^t \cap \Lambda_m^t = \emptyset$ and line 9 $|\Lambda_m^t| = |\Lambda_0^t| + \frac{m \times n_b^t}{N}$

---

**Algorithm 5** Adaptive Dynamic Sampling for Unified Patterns

---

**Input** : Initial data $\tilde{\mathbf{y}}_0 \in \mathbb{C}^{n \times n_c \times n_f}$, Acceleration $R$, Sample Space $\Omega$
**Output:** Acquired $k$-space data $\tilde{\mathbf{y}}$ for $R$
$n_a \leftarrow |\Omega|$;
$n_b \leftarrow \frac{n_a}{R} - |\Lambda_0|$ ;                                  // Calculate total budget
**for** $m \leftarrow 1$ *to* $N$ **do**
    $\mathbf{p}_m \leftarrow \mathcal{M}_{\boldsymbol{\psi}_m} \circ \mathcal{E}_{\boldsymbol{\theta}_m}(\tilde{\mathbf{y}}_{m-1})$ ;         // Produce adaptive probabilities $\mathbf{p}_m \in \mathbb{R}^{n_a}$
    **for** *each* $i \in \left( \bigcup_{j=0}^{m-1} \Lambda_j \right)$ **do**
        $(\mathbf{p}_m)_i \leftarrow 0$ ;                              // Zero-out already sampled indices
    **end**
    $\mathbf{p}_m \leftarrow \text{Softplus}(\mathbf{p}_m)$ ;                                      // Apply Softplus
    $\mathbf{p}_m \leftarrow \text{Rescale}(\mathbf{p}_m; \texttt{budget} = n_b/N)$ ;      // Rescale such that $\mathbb{E}(\mathbf{p}_m) = \frac{n_b}{N \times n_a}$
    $\Lambda_m \leftarrow \text{STE}(\mathbf{p}_m)$ ;                              // Produce adaptive sampling
    $\Lambda_m \leftarrow (\Lambda_m, \cdots, \Lambda_m)$ ;                    // Apply same pattern across frames
    $\mathbf{z}_m \leftarrow \mathbf{U}_{\Lambda_m} \mathbf{y}$ ;                              // Acquire new $k$-space on $\Lambda_m$
    $\tilde{\mathbf{y}}_m \leftarrow \tilde{\mathbf{y}}_{m-1} + \mathbf{z}_m = \mathbf{U}_{\cup_{j=0}^{m} \Lambda_j} \mathbf{y}$ ;         // Aggregate with previous data $\tilde{\mathbf{y}}_{m-1}$
**end**
$\tilde{\mathbf{y}} \leftarrow \tilde{\mathbf{y}}_N$;
 **return** $\tilde{\mathbf{y}}$
 // Note: Lines 5-6 ensure $\Lambda_{m-1} \cap \Lambda_m = \emptyset$ and line 8 $|\Lambda_m| = |\Lambda_0| + \frac{m \times n_b}{N}$

---

## A.4. End-to-end Adaptive Dynamic Subsampling and Reconstruction

---
**Algorithm 6** End-to-end Adaptive Dynamic Sampling and Reconstruction
---
**Input**  : Initial data $\tilde{\mathbf{y}}_{\Lambda_0}$, ACS data $\tilde{\mathbf{y}}_{\Lambda_{\mathrm{acs}}} : \Lambda_{\mathrm{acs}} \subseteq \Lambda_0$, Acceleration factor $R$
**Output:** Reconstructed image $\hat{\mathbf{x}}$
**for** $k \leftarrow 1$ *to* $n_c$ **do**
$\quad$ **for** $t \leftarrow 1$ *to* $n_f$ **do**
$\quad\quad$ $\mathbf{S}_t^k \leftarrow \mathcal{S}_{\boldsymbol{\omega}}(\tilde{\mathbf{y}}_{\Lambda_{\mathrm{acs}}^t}^k)$ ;         `// Estimate sensitivity maps`
$\quad$ **end**
**end**
$\tilde{\mathbf{y}} \leftarrow \mathrm{ADS}_{\boldsymbol{\psi},\boldsymbol{\theta}}(\tilde{\mathbf{y}}_{\Lambda_0}; \mathbf{S}, R)$ ;   `// Adaptively sample $k$-space based on $\tilde{\mathbf{y}}_{\Lambda_0}$ and $R$`
$\hat{\mathbf{x}} \leftarrow \mathcal{R}_{\boldsymbol{\phi}}(\tilde{\mathbf{y}}]; \mathbf{S})$ ;       `// Reconstruct dynamic image`
**return** $\hat{\mathbf{x}}$
---

# Appendix B. Additional Information

## B.1. Methods

### B.1.1. STRAIGHT-THROUGH ESTIMATOR

In the ADS component of our proposed method, we employ a straight-through estimator (STE) to binarize the predicted probabilities, which have been rescaled to the acceleration factor and zeroed out at already sampled locations. This is key for generating a binary mask from continuous probability values. The STE employs random uniform sampling in the forward pass, where each predicted probability is compared against a randomly drawn value from a uniform distribution (see Algorithm 2). This step is crucial for backpropagation because it allows the estimator to handle the non-differentiable nature of binarization. If a deterministic method like the top-k operator was used, the hard thresholding would result in non-differentiable operations, blocking gradient flow. By using random sampling, we create a smoother decision boundary that the STE can approximate during the backward pass with a sigmoid function.

Note that this stochasticity not only introduces randomness during training but also allows the model to better simulate real-world scenarios where decisions are not always deterministic. Additionally, the stochastic nature of binarization is used during inference to maintain variability in the binary decision-making process.

In the backward pass (see Algorithm 3), the STE approximates the non-differentiable binarization step with a sigmoid function that has a slope of 10, enabling gradients to propagate through the discrete sampling operation. This allows for end-to-end training via backpropagation. Without this smooth approximation, gradients would vanish due to the hard thresholding, preventing effective learning.

### B.1.2. HANDLING COMPLEX-VALUED OPERATIONS

Complex-valued data, including images, $k$-space data, and sensitivity maps, were decomposed into their real and imaginary components, then stacked along the channel dimension (with size 2) for processing by real-valued model weights. As a result, all model weights were real-valued. Operations like the Fourier transform and its inverse were applied by temporarily converting the data back to its complex form when required.

## B.2. Experimental Setup

### B.2.1. DATASET

As outlined in Section 3.1, we used the cine CMRxRecon challenge 2023 dataset (Wang et al., 2023, 2021). Specifically, the data were acquired using a 3T MRI scanner with a 'TrueFISP' readout. The dataset includes short-axis (SA), two, three and four-chamber long-axis (LA) views. Each scan consists of fully-sampled (ECG-triggered acquisition) multi-coil acquisitions ($n_c = 10$) with 3-12 dynamic (2D + time) slices. The cardiac cycle was segmented into $n_f = 12$ temporal phases (referred to as frames in the paper), with a temporal resolution of 50 ms. The spatial resolution was $2.0 \times 2.0$ mm$^2$, with a slice thickness of 8.0 mm and a slice gap of 4.0 mm.

### B.2.2. DATA PREPROCESSING

As detailed in Section 2.3, each MLP component $\mathcal{M}_{\psi_m}$ within each cascade receives a flattened image as input, which requires a fixed input shape due to the MLP's fixed number of features. To achieve this, all data were center zero-padded to match the largest spatial size in the dataset, i.e., $(n_1, n_2) = (512, 246)$. This process involved transforming the multi-coil *k*-space data into the image domain using the inverse Fourier transform, applying center zero-padding, and then transforming the data back into the frequency domain via the Fourier transform. Data were normalized using the $99.5^{th}$ percentile value of the magnitude of the fully sampled autocalibration signal for each case:

$$s = \text{quantile}_{99.5}(|\mathbf{y}_{\Lambda_{\text{acs}}}|).$$

### B.2.3. SAMPLING SCHEMES

In our experimental setup we consider predetermined or random schemes for comparison to our proposed methodology. Following the algorithms available in the literature (Yiasemis et al., 2024c) we specifically consider:

- Equispaced (1D/line): Lines selected at fixed intervals based on the desired acceleration, with a randomly selected offset.
- Random (1D/line): Lines selected from a uniform distribution up to the desired acceleration.
- Gaussian 1D (1D/line): Lines selected from a 1D Gaussian distribution with mean $\mu = n_2/2$ and standard deviation $\sigma = 4\sqrt{\mu}$.
- Gaussian 2D (2D/point): Samples drawn from a 2D Gaussian distribution with mean $\boldsymbol{\mu} = (n_1/2, n_2/2)$ and standard deviation $\boldsymbol{\Sigma} = 4\mathbf{I}\sqrt{\boldsymbol{\mu}}^T$.
- Radial (2D/point): Samples selected in a radial fashion on the Cartesian grid using the CIRCUS method.

For the above, in frame-specific experiments a distinct (arbitrary random seed) pattern from a scheme was generated per frame, whereas for unified an identical scheme was applied to all frames. For frame-specific experiments, within each setup, a distinct random pattern was generated per frame, while unified sampling experiments used the same pattern for all frames. We also evaluated *k*t schemes that generate dynamic sampling with temporal interleaving, avoiding repeated sampling in adjacent frames:

- *k*t-Equispaced
- *k*t-Gaussian 1D
- *k*t-Radial

During training, arbitrary patterns were generated without fixed seeds to maximize model exposure to varied data. At inference, the seed for generating patterns was fixed for each scan/patient to ensure consistency (e.g. during validation).

### B.2.4. LOSS FUNCTION DEFINITIONS

This study utilizes multiple loss components calculated either in the image domain or the frequency domain and are derived from established literature. The definitions of these components are as follows:

- Structural Similarity Index Measure (SSIM) Loss

$$\mathcal{L}_{\text{SSIM}} := 1 - \text{SSIM}, \quad \text{SSIM}(\mathbf{z}, \mathbf{w}) = \frac{1}{N} \sum_{i=1}^{N} \frac{(2\mu_{\mathbf{z}_i}\mu_{\mathbf{w}_i} + \gamma_1)(2\sigma_{\mathbf{z}_i\mathbf{w}_i} + \gamma_2)}{(\mu_{\mathbf{z}_i}^2 + \mu_{\mathbf{w}_i}^2 + \gamma_1)(\sigma_{\mathbf{z}_i}^2 + \sigma_{\mathbf{w}_i}^2 + \gamma_2)},$$

where $\mathbf{z}_i, \mathbf{w}_i, i = 1, ..., N$ represent $7 \times 7$ square windows of $\mathbf{z}, \mathbf{w}$, respectively, and $\gamma_1 = 0.01$, $\gamma_1 = 0.03$. Additionally, $\mu_{\mathbf{z}_i}$, $\mu_{\mathbf{w}_i}$ denote the means of each window, $\sigma_{\mathbf{z}_i}$ and $\sigma_{\mathbf{w}_i}$ represent the corresponding standard deviations. Lastly, $\sigma_{\mathbf{z}_i\mathbf{w}_i}$ represents the covariance between $\mathbf{z}_i$ and $\mathbf{w}_i$.
- Structural Similarity Index Measure 3D (SSIM3D) Loss

$$\mathcal{L}_{\text{SSIM3D}} := 1 - \text{SSIM3D},$$

where SSIM3D follows the same definition as SSIM, but replacing the $7 \times 7$ windows with cubic windows $7 \times 7 \times 7$.
- Mean Average Error ($L_1$) Loss

$$\mathcal{L}_1(\mathbf{z}, \mathbf{w}) = ||\mathbf{z} - \mathbf{w}||_1 = \sum_{i=1}^{n} |z_i - w_i|$$

- High Frequency Error Norm (HFEN)

$$\mathcal{L}_{\text{HFEN}} := \text{HFEN}, \quad \text{HFEN}(\mathbf{z}, \mathbf{w}) = \frac{||\mathcal{G}(\mathbf{z}) - \mathcal{G}(\mathbf{w})||_1}{||\mathcal{G}(\mathbf{w})||_1},$$

where $\mathcal{G}$ is a Laplacian-of-Gaussian filter with kernel of size $15 \times 15$ and with a standard deviation of 2.5.
- Normalized Mean Average Error (NMAE)

$$\mathcal{L}_{\text{NMAE}} := \text{NMAE}, \quad \text{NMAE}(\mathbf{z}, \mathbf{w}) = \frac{||\mathbf{z} - \mathbf{w}||_1}{||\mathbf{z}||_1} = \frac{\sum_{i=1}^{n} |z_i - w_i|}{\sum_{i=1}^{n} |z_i|}.$$

### B.2.5. SELECTION OF OPTIMAL MODEL CHECKPOINTS

Results were obtained using the best-performing model checkpoints, selected based on validation set performance using the SSIM metric.

### B.2.6. SIGNIFICANCE TESTING

In our study, we used the almost stochastic order (ASO) test (Dror et al., 2019) with a significance level of $\alpha = 0.05$ to compare reconstruction metrics between models due to its robustness in handling complex data distributions. Traditional parametric tests, such as the t-test, assume that the differences between models follow a normal distribution and have equal variances. However, these assumptions are not always valid in deep learning contexts where data distributions can be highly irregular and performance metrics can be influenced by various stochastic factors. ASO does not rely on such assumptions. Instead, it evaluates the degree to which one distribution stochastically dominates another, providing a more reliable assessment of significance when comparing models. This approach is particularly suitable for deep learning applications where performance metrics can be non-normally distributed and vary across experiments.

### B.3. Reconstruction Model Robustness Experiments

To assess the robustness of our end-to-end pipeline, we repeated the comparative studies using a state-of-the-art reconstruction model, specifically the Model-based neural network with enhanced deep learned regularizers (MEDL-Net) (Qiao et al., 2023), instead of vSHARP. Below, we provide details on the optimization process and architectural design.

#### B.3.1. Optimization

The models were developed in PyTorch (Paszke et al., 2019), following the same training scheme as our main experiments. We used the Adam optimizer with an initial learning rate of $1 \times 10^{-3}$, which increased linearly to $3 \times 10^{-3}$ over 2,000 iterations and subsequently decreased by 20% every 10,000 iterations, over a total of 52,000 iterations. Experiments were conducted on single NVIDIA A6000 or A100 GPUs with a batch size of 1.

For training, we adopted the loss function proposed by the authors in the original publication (Qiao et al., 2023), computed exclusively in the image domain:

$$\mathcal{L} = \sum_{j=1}^{T} w_j \sum_{t=1}^{n_f} \mathcal{L}_{\text{MSE}}(\hat{\mathbf{x}}_t^{(j)}, \mathbf{x}_t^*), \quad w_j = \left\{ \begin{array}{ll} 0.1, & j = 1, \cdots, T-1 \\ 1, & j = T \end{array} \right. .$$

where $\{\hat{\mathbf{x}}^{(j)}\}_{j=1}^{T}$ denotes the predicted dynamic images from MEDL's unrolled steps, and $\mathcal{L}_{\text{MSE}}$ the mean squared error loss function.

#### B.3.2. Hyperparameter Settings

For sensitivity map estimation, we used an SMP configuration identical to that in our adaptive sampling experiments. Similarly, in our adaptive sampling experiments, the ADS model module was configured identically to that in the vSHARP-based reconstruction experiments (see Section II.F.3 of the main paper). For MEDL, we employed the default hyperparameters as specified in the corresponding publication (Qiao et al., 2023).

## Appendix C. Quantitative Results

### C.1. Supplementary Results

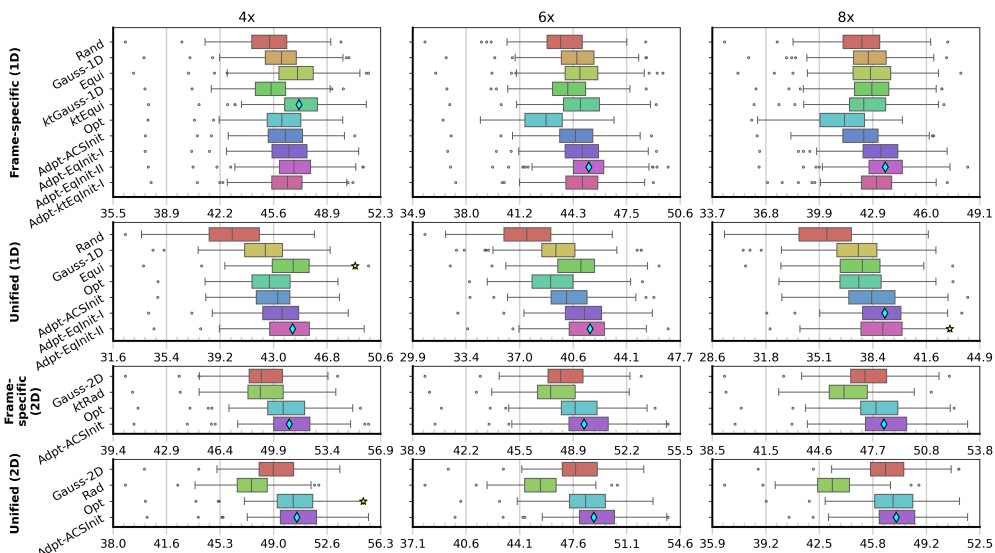

Figure S1: PSNR metrics across all experimental settings and setups. Diamonds ($\Diamond$) on the box-plot median indicate the average best methods. A star ($\star$) on the upper whisker indicates non-significance in comparison to the average best method.

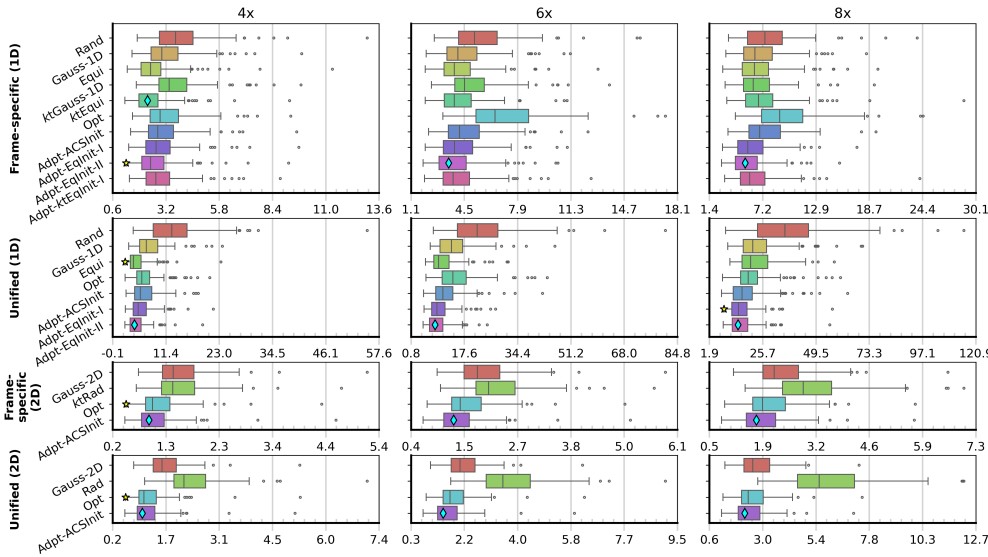

Figure S2: NMSE ($\times 100$) metrics across all experimental settings and setups. Diamonds ($\Diamond$) on the box-plot median indicate the average best methods. A star ($\star$) on the lower whisker indicates non-significance in comparison to the average best method.

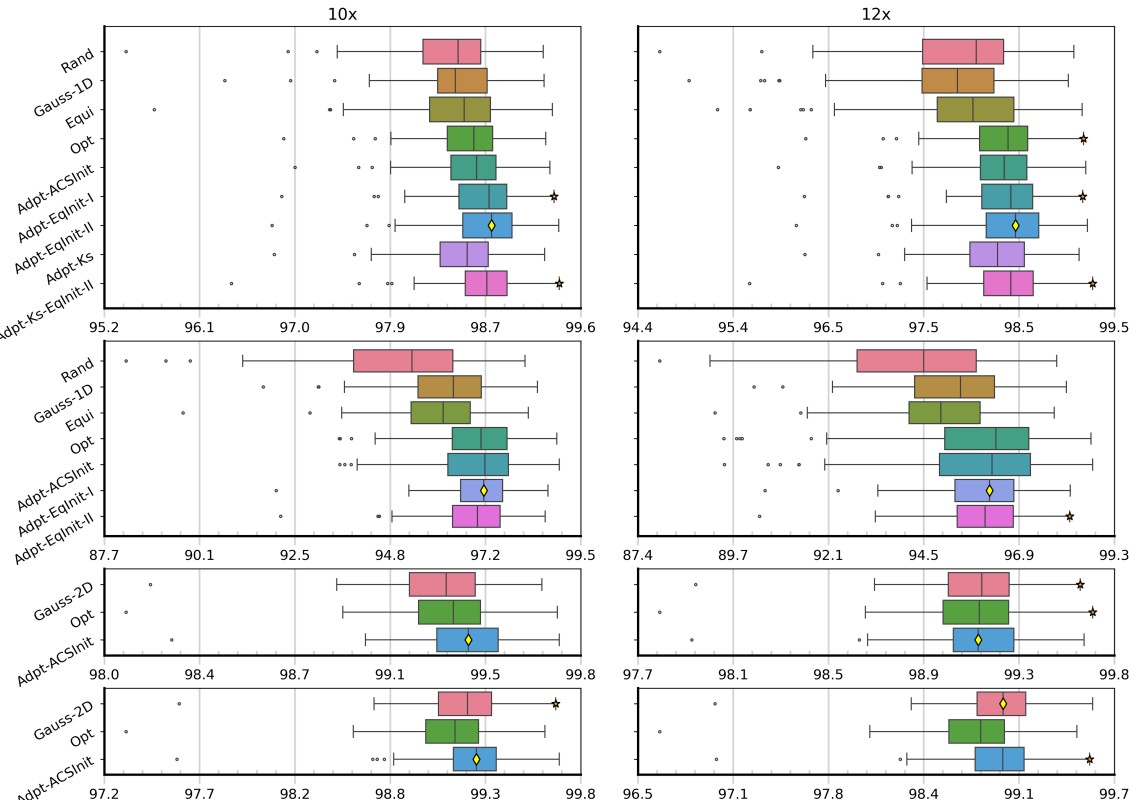

Figure S3: SSIM ($\times100$) metrics across all experimental settings in the main experiments evaluated at unseen during training accelerations ($10\times$ and $12\times$). Diamonds ($\Diamond$) on the box-plot median indicate the average best methods. A star ($\star$) on the lower whisker indicates non-significance in comparison to the average best method.

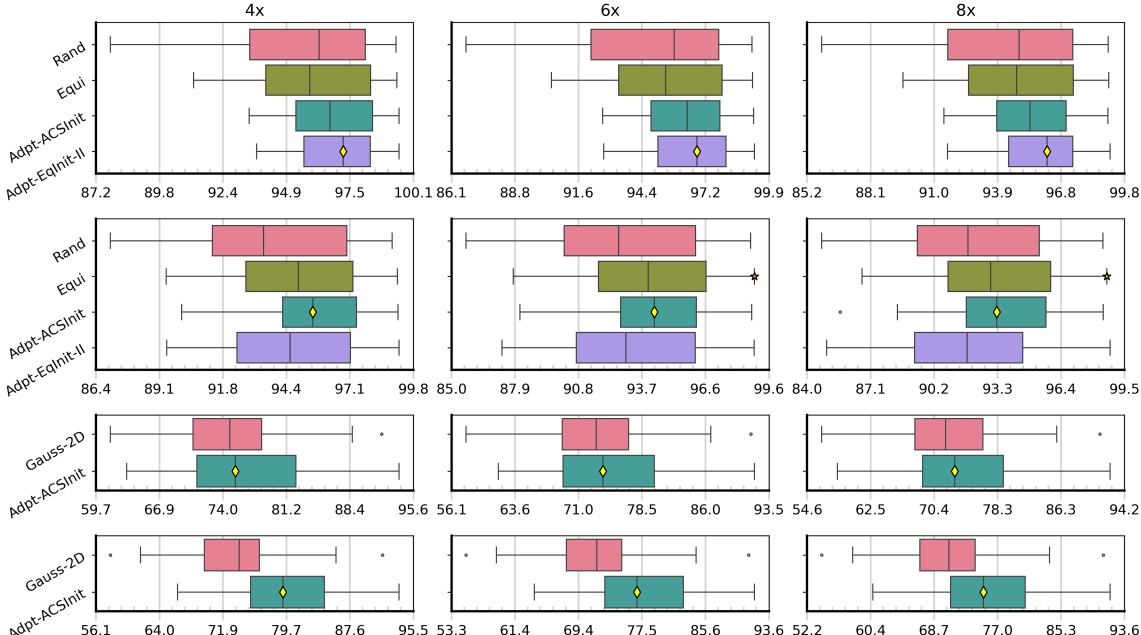

Figure S4: SSIM ($\times 100$) metrics across all experimental settings and setups evaluated on an external dataset, unseen during training. Diamonds ($\Diamond$) on the box-plot median indicate the average best methods. A star ($\star$) on the lower whisker indicates non-significance in comparison to the average best method.

## C.2. Inference Times

| Frame-specific (1D) | |
|---|---|
| **Method** | **Time (s)** |
| Rand | $10.59 \pm 5.46$ |
| Gauss-1D | $10.63 \pm 5.42$ |
| Equi | $10.65 \pm 5.47$ |
| *kt*Gauss-1D | $10.58 \pm 5.45$ |
| *kt*Equi | $10.62 \pm 5.43$ |
| Opt | $10.74 \pm 5.54$ |
| Adpt-ACSInit | $11.27 \pm 5.77$ |
| Adpt-EqInit-I | $11.05 \pm 5.71$ |
| Adpt-EqInit-II | $11.33 \pm 5.81$ |
| Adpt-*kt*EqInit-I | $11.24 \pm 5.81$ |

| Unified (1D) | |
|---|---|
| **Method** | **Time (s)** |
| Rand | $11.07 \pm 5.74$ |
| Gauss-1D | $10.66 \pm 5.47$ |
| Equi | $10.96 \pm 5.65$ |
| Opt | $10.52 \pm 5.42$ |
| Adpt-ACSInit | $11.10 \pm 5.73$ |
| Adpt-EqInit-I | $10.92 \pm 5.58$ |
| Adpt-EqInit-II | $10.71 \pm 5.50$ |

| Frame-specific (2D) | |
|---|---|
| **Method** | **Time (s)** |
| Gauss-2D | $10.70 \pm 5.46$ |
| *kt*Rad | $10.74 \pm 5.48$ |
| Opt | $10.67 \pm 5.47$ |
| Adpt-ACSInit | $11.17 \pm 5.73$ |

| Unified (2D) | |
|---|---|
| **Method** | **Time (s)** |
| Gauss-2D | $10.92 \pm 5.60$ |
| Rad | $11.07 \pm 5.68$ |
| Opt | $10.62 \pm 5.48$ |
| Adpt-ACSInit | $11.05 \pm 5.71$ |

Table S1: Average inference times (in seconds) for all sampling and reconstruction configurations at $R = 4$, measured on a single NVIDIA A100 GPU. Each table reports the mean and standard deviation of total runtime per test volume, including both the sampling step (adaptive or non-adaptive) and the subsequent reconstruction.

Inference runtime is reported in Table S1 for all configurations. Given that each 4D volume in our dataset comprises $n_f = 12$ frames and between 3 to 12 slices (see Appendix B.2.1), we estimate a conservative upper bound on per-frame sampling prediction time in frame-specific settings to be approximately $\frac{1}{12 \times 3} = 27$ milliseconds per frame prediction, based on the observed $< 1$ second overhead.

## C.3. Ablation Results

Table S2: Average results for using a single cascade ($N = 1$). **Bold** numbers indicate better performance than using two cascades ($N = 2$).

| Method | | 4× | | | 6× | | | 8× | | |
|---|---|---|---|---|---|---|---|---|---|---|
| | | SSIM | pSNR | NMSE | SSIM | pSNR | NMSE | SSIM | pSNR | NMSE |
| Frame-specific | Adpt-ACSInit-N1 | **0.9879** | **46.43** | **0.0030** | **0.9848** | **45.11** | **0.0041** | **0.9815** | **43.89** | **0.0053** |
| (1D) | Adpt-EqInit-II-N1 | 0.9878 | 46.62 | 0.0029 | 0.9841 | 44.97 | 0.0042 | 0.9793 | 43.45 | 0.0059 |
| Frame-specific | Adpt-ACSInit-N1 | 0.9937 | 50.92 | 0.0011 | 0.9922 | **49.75** | **0.0014** | 0.9902 | **48.54** | **0.0018** |
| (2D) | | | | | | | | | | |
| Unified | Adpt-ACSInit-N1 | 0.9744 | 42.61 | 0.0072 | 0.9633 | 39.89 | 0.0134 | 0.9503 | 37.97 | 0.0201 |
| (1D) | Adpt-EqInit-II-N1 | 0.9753 | 42.85 | 0.0070 | 0.9628 | 40.31 | 0.0124 | 0.9452 | 37.92 | 0.0212 |
| Unified | Adpt-ACSInit-N1 | 0.9930 | 50.45 | 0.0012 | 0.9910 | 48.99 | 0.0017 | 0.9880 | 47.33 | 0.0024 |
| (2D) | | | | | | | | | | |

Table S3: Average results for allowing the adaptive sampler to non uniformly allocate the sampling budget across time-frames.

| Method | | 4× | | | 6× | | | 8× | | |
|---|---|---|---|---|---|---|---|---|---|---|
| | | SSIM | pSNR | NMSE | SSIM | pSNR | NMSE | SSIM | pSNR | NMSE |
| Frame-specific | Adpt-NU | 0.9881 | 46.32 | 0.0031 | 0.9833 | 44.34 | 0.0048 | 0.9755 | 41.91 | 0.0085 |
| (1D) | Adpt-NU-N1 | 0.9873 | 46.27 | 0.0031 | 0.9830 | 44.58 | 0.0047 | 0.9769 | 42.46 | 0.0075 |

## C.4. Reconstruction Model Robustness Results

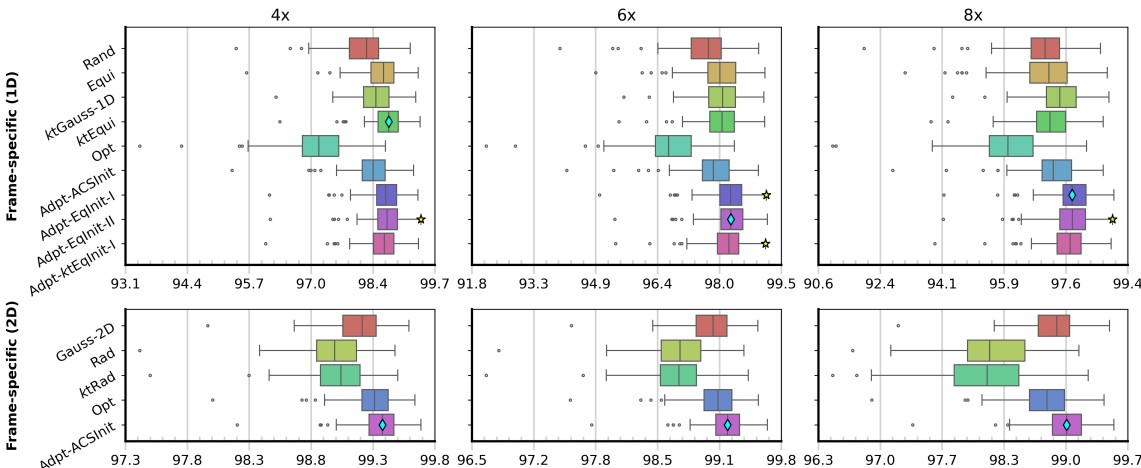

Figure S5: SSIM ($\times 100$) metrics across all experimental settings and setups using MEDL as a reconstruction network. Diamonds ($\diamond$) on the box-plot median indicate the average best methods. A star ($\star$) on the lower whisker indicates non-significance in comparison to the average best method.

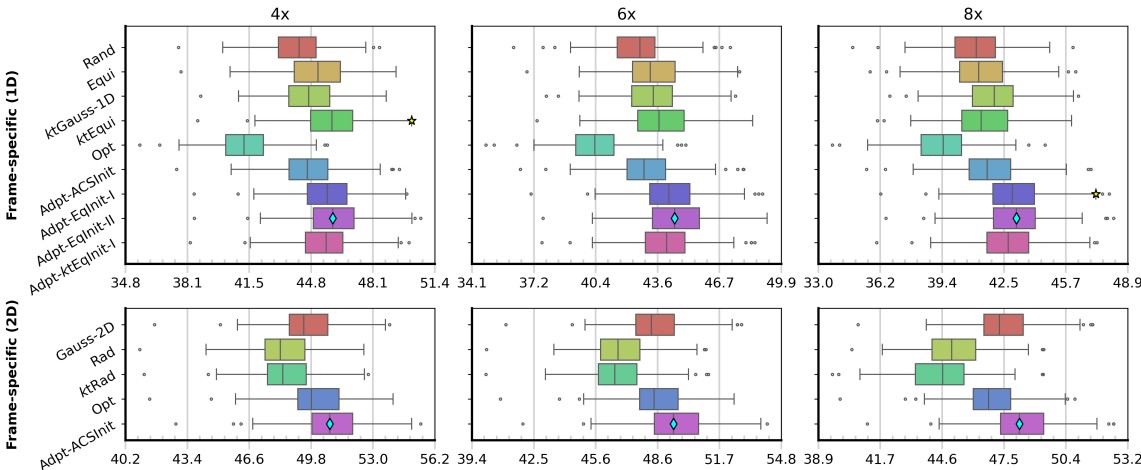

Figure S6: PSNR metrics across all experimental settings and setups using MEDL as a reconstruction network. Diamonds ($\diamond$) on the box-plot median indicate the average best methods. A star ($\star$) on the lower whisker indicates non-significance in comparison to the average best method.

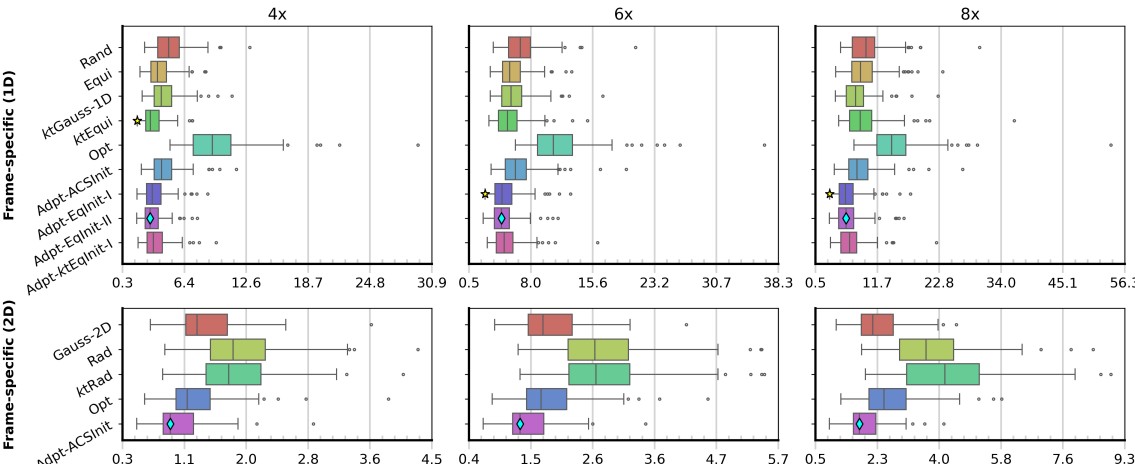

Figure S7: NMSE ($\times 100$) metrics across all experimental settings and setups using MEDL as a reconstruction network. Diamonds ($\Diamond$) on the box-plot median indicate the average best methods. A star ($\star$) on the lower whisker indicates non-significance in comparison to the average best method.

## Appendix D. Qualitative Results

In this section we illustrate examples of produced subsampling patterns and reconstructions across all experiments (frame-specific or unified, 1D or 2D sampling) and setups (fixed or learned). Black lines or points indicate fixed or initial pattern (applicable only to learned if initial), and red lines or points indicate learned pattern. Cyan boxes mark the SSIM values.

### D.1. Phase-Specific Experiments

#### D.1.1. SUBSAMPLING PATTERNS

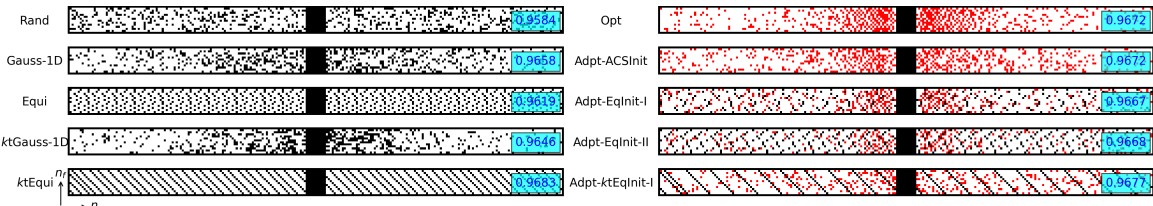

Figure S8: Example of patterns from the test set across setups for frame-specific sampling at $R = 4$.



Figure S9: Example of patterns from the test set across setups for frame-specific sampling at $R = 6$.

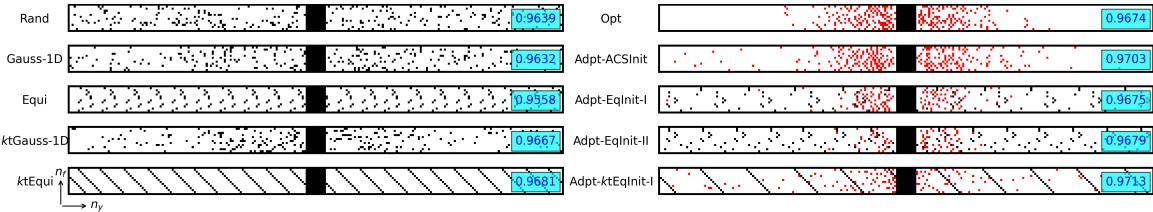

Figure S10: Example of patterns from the test set across setups for frame-specific sampling at $R = 8$.

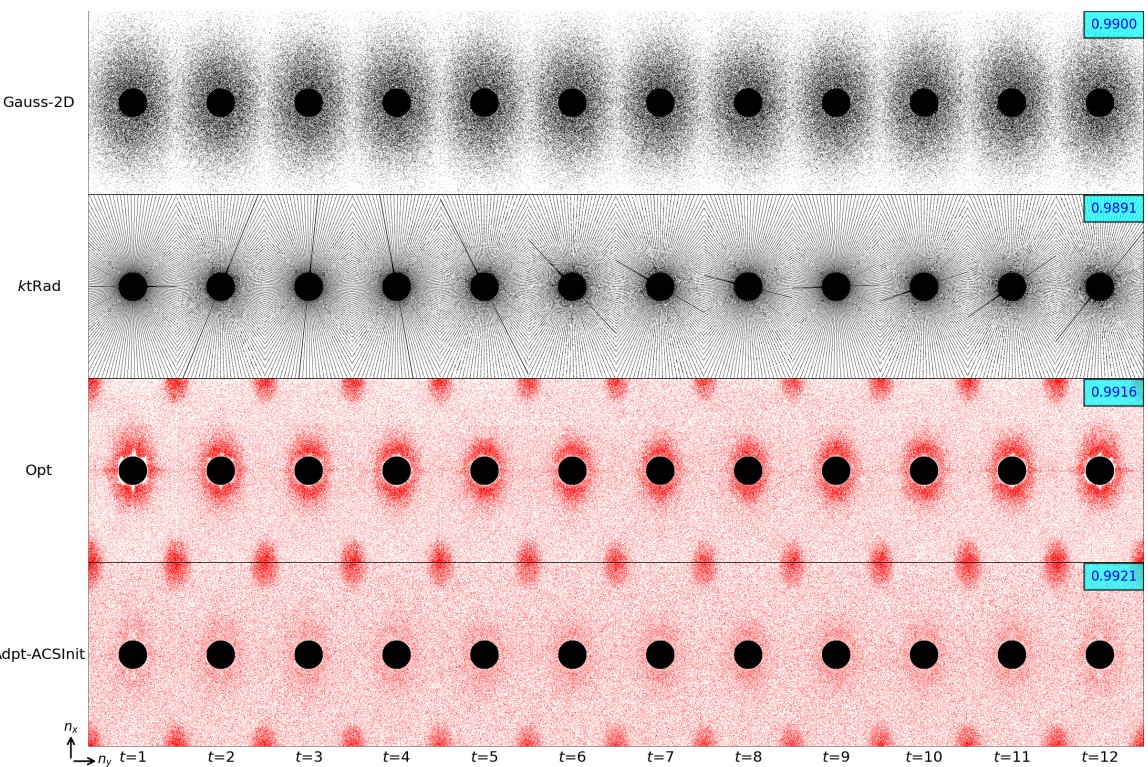

Figure S11: Example of patterns from the test set across setups for frame-specific 2D sampling at $R = 4$.

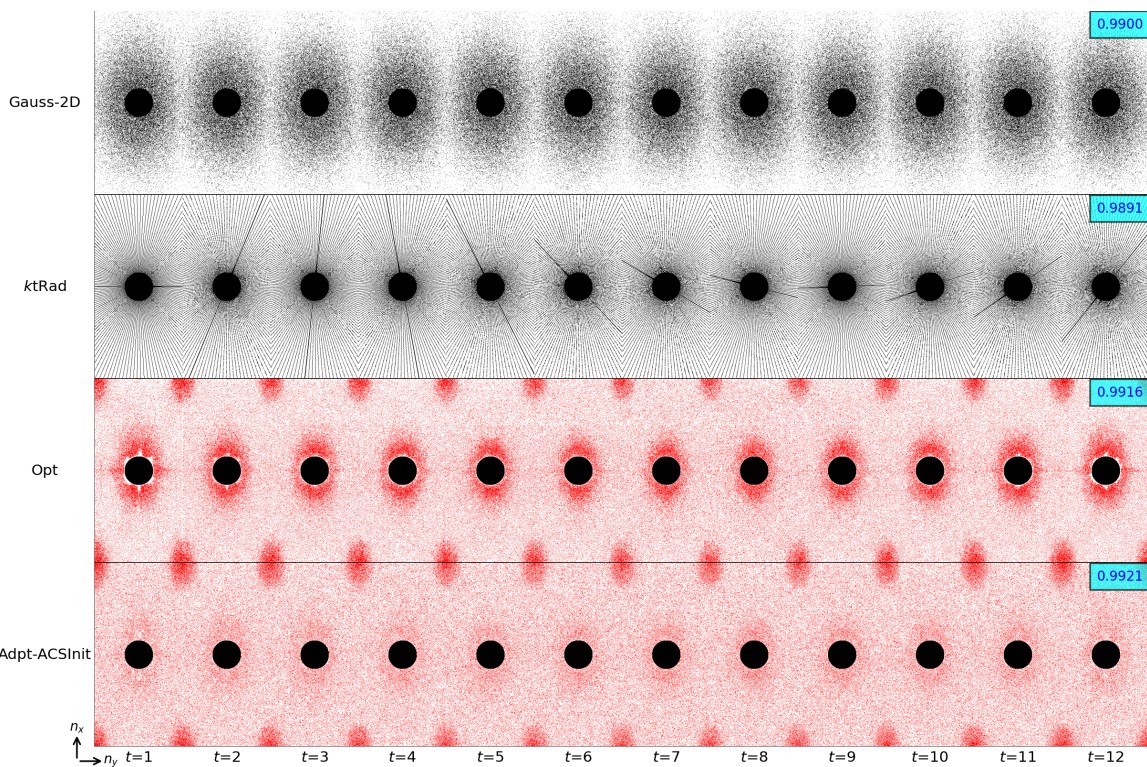

Figure S12: Example of patterns from the test set across setups for frame-specific 2D sampling at $R = 6$.

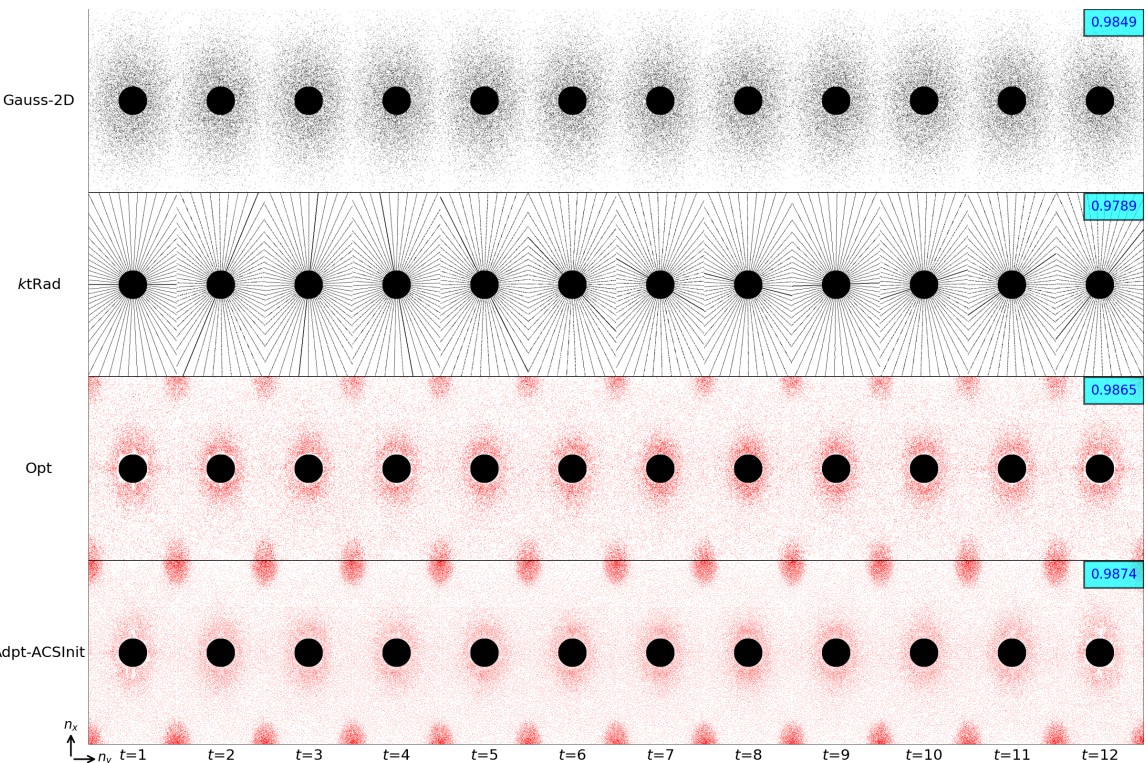

Figure S13: Example of patterns from the test set across setups for frame-specific 2D sampling at $R = 8$.

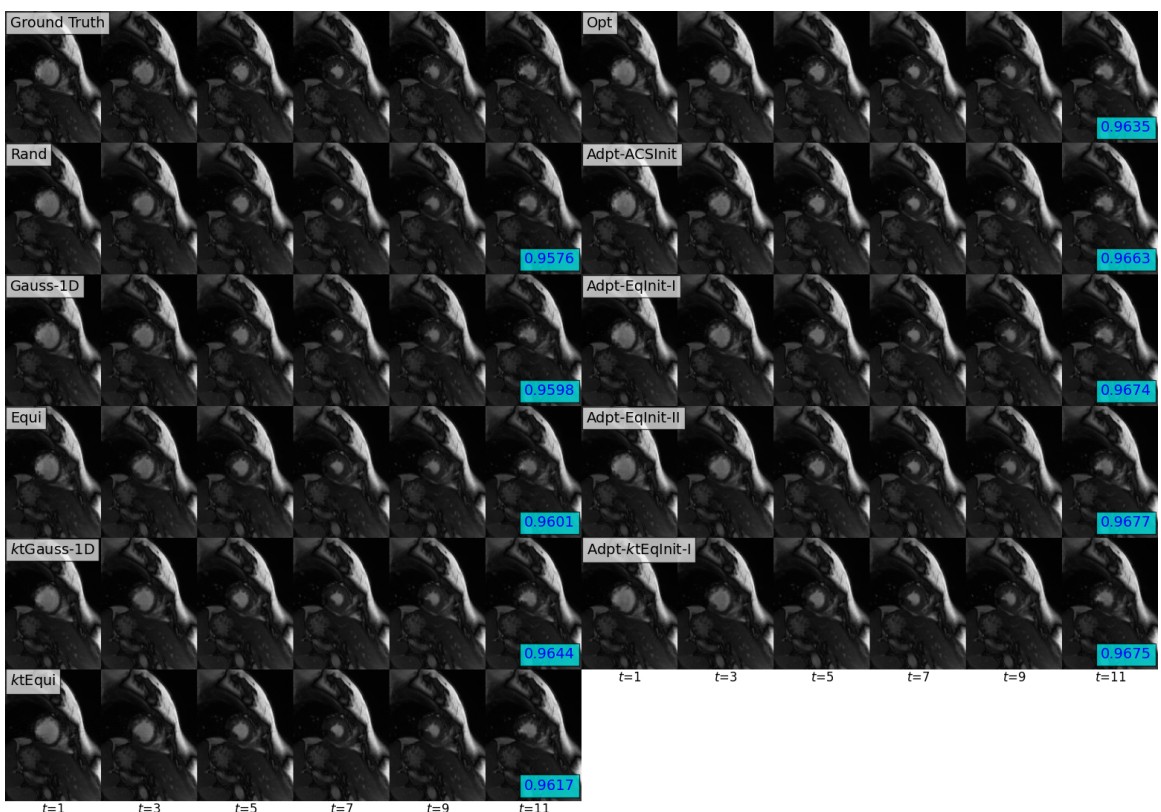

Figure S14: Example of reconstructions from the test set across setups for frame-specific 1D sampling at $R = 8$.

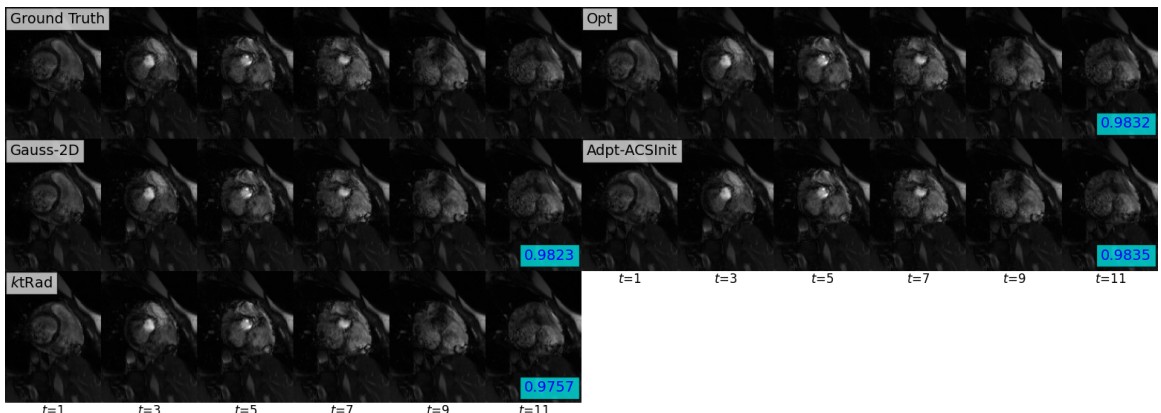

Figure S15: Example of reconstructions from the test set across setups for frame-specific 2D sampling at $R = 8$.

## D.2. Unified Experiments

### D.2.1. Subsampling Patterns

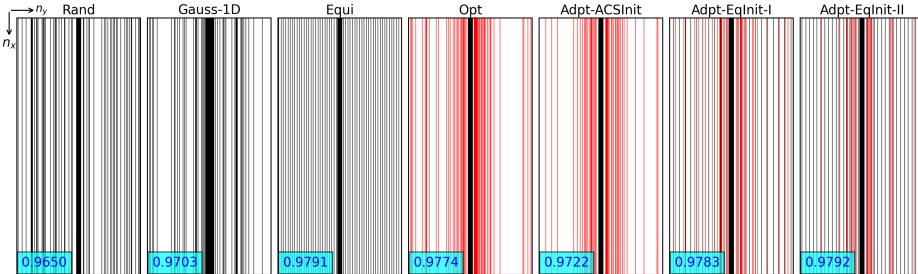

Figure S16: Example of patterns from the test set across setups for unified sampling at $R = 4$.

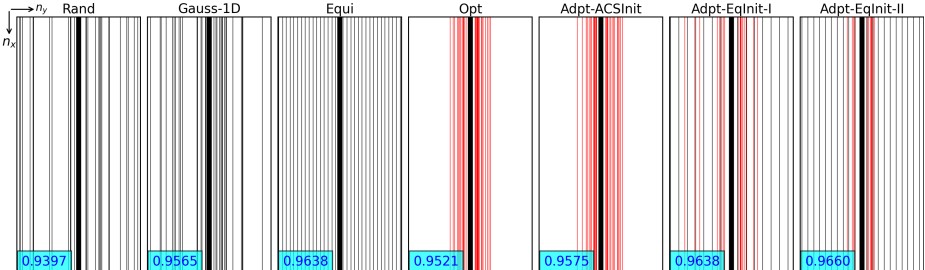

Figure S17: Example of patterns from the test set across setups for unified sampling at $R = 6$.

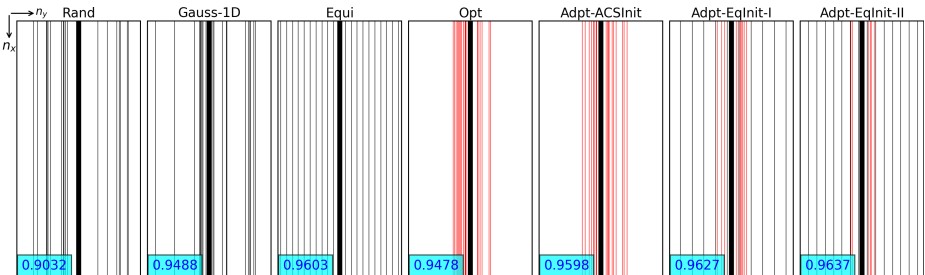

Figure S18: Example of patterns from the test set across setups for unified sampling at $R = 8$.

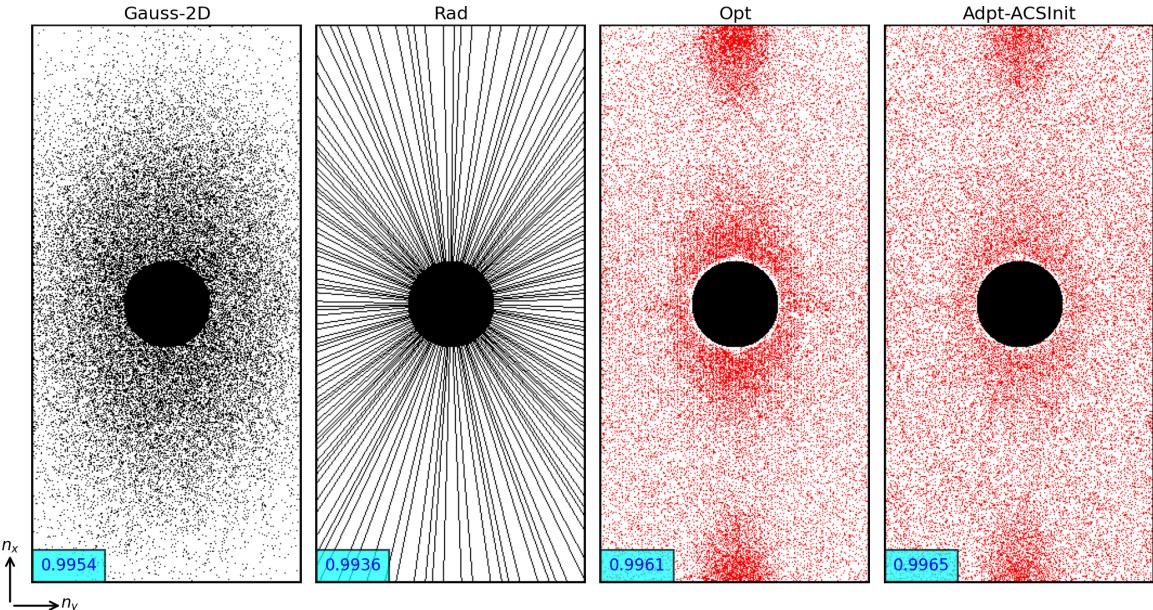

Figure S19: Example of patterns from the test set across setups for unified 2D sampling at $R = 4$.

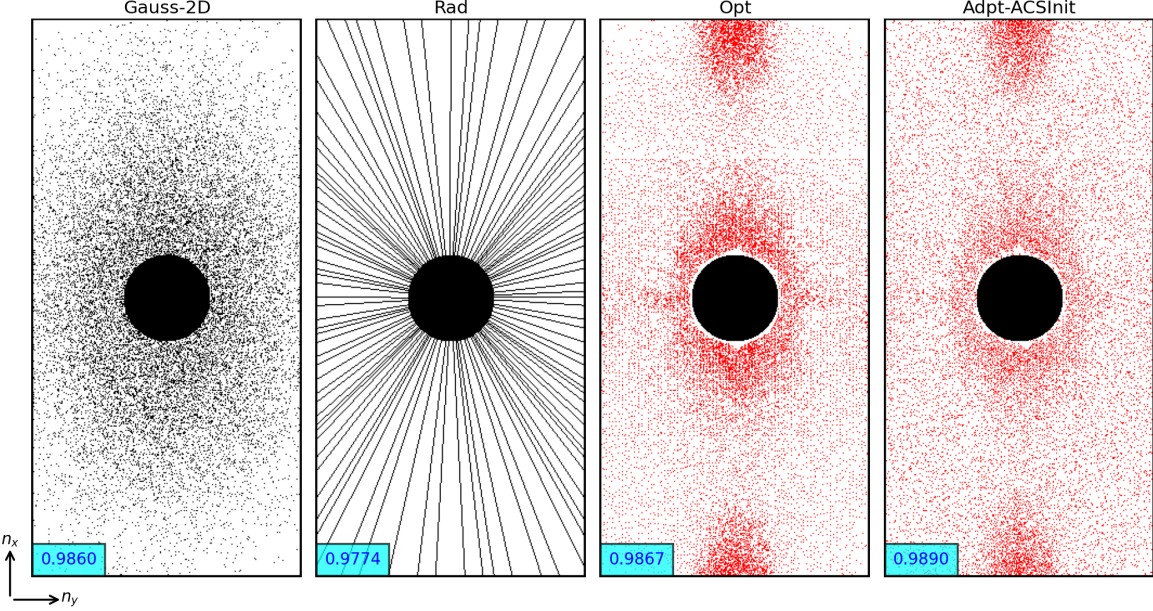

Figure S20: Example of patterns from the test set across setups for unified 2D sampling at $R = 4$.

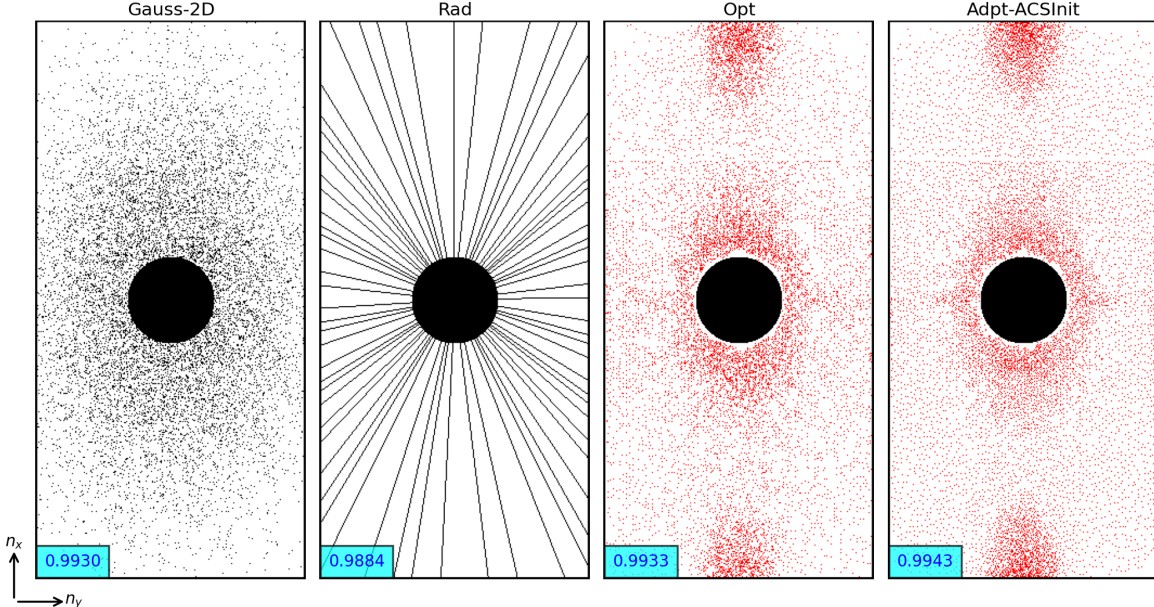

Figure S21: Example of patterns from the test set across setups for unified 2D sampling at $R = 8$.

## D.2.2. Reconstructions

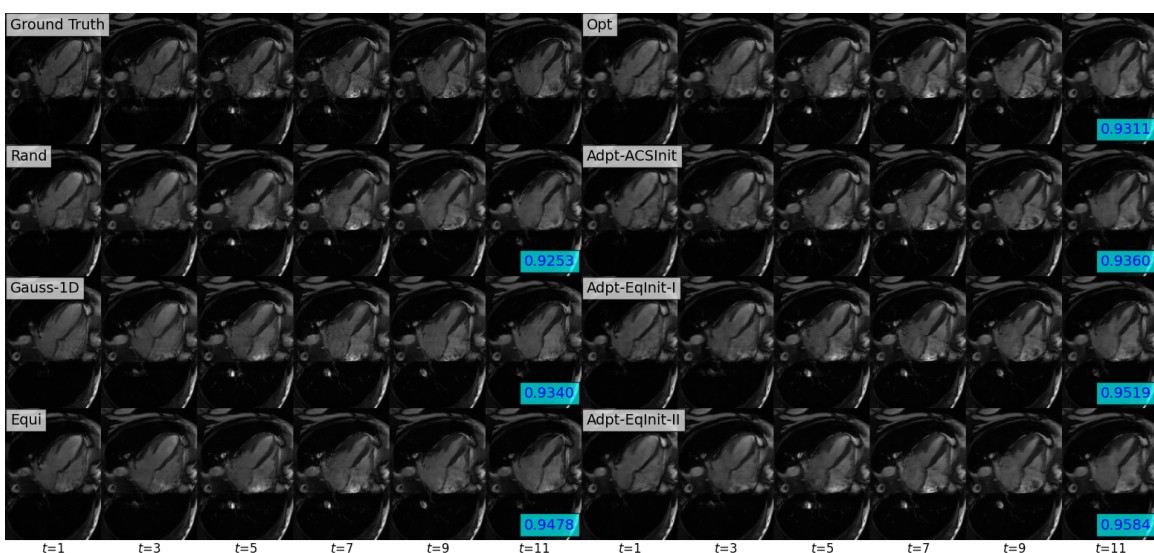

Figure S22: Example of reconstructions from the test set across setups for unified 1D sampling at $R = 8$.

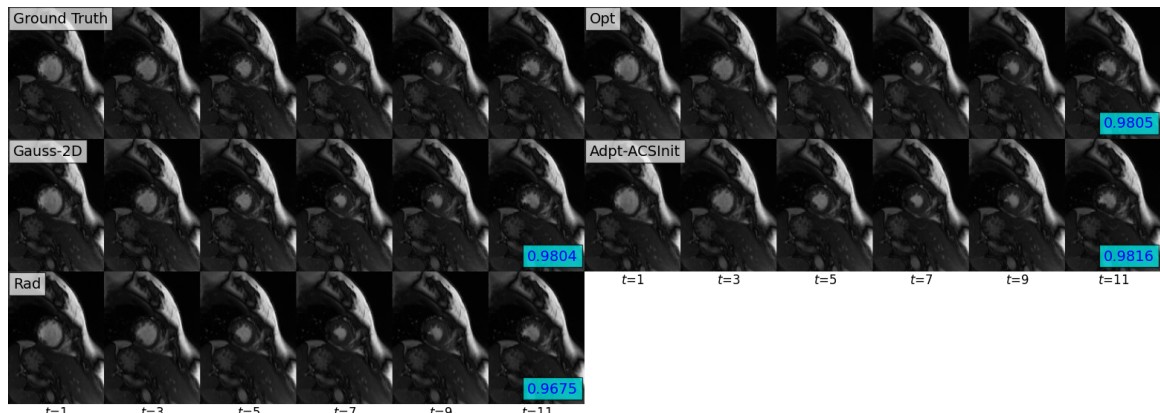

Figure S23: Example of reconstructions from the test set across setups for unified 2D sampling at $R = 8$.

