# OpenReview forum: "End-to-End Co-Optimization of Adaptive $k$-space Sampling and Reconstruction for Dynamic MRI"
_MIDL.io/2026/Conference — MIDL 2026 Poster_

### Official Review · Reviewer_JQeN · 2025-12-18

**Confidence:** 5
**Preliminary Rating:** 4
**Final Rating:** 4

**Summary:**

The paper proposes an end-to-end adaptive k-space sampling method for dynamic MRI reconstruction. The method is validated using the publicly available CMRxRecon dataset. The authors compare their method with several other approaches, such as random, predetermined, and non-adaptive optimized learning schemes. The work is significant because it can be used to accelerate dynamic MRI acquisitions further.

**Strengths:**

- The paper is well-written and relatively easy to follow, though a lot of details were pushed to the appendix of the paper.
- The paper presents a very detailed comparison against other types of sampling schemes.
- The work uses public data, and the code will be made publicly available.

**Weaknesses:**

Although the paper is very detailed, I missed a more thorough evaluation of the results from either a qualitative or downstream-task perspective.  Looking at Figure 5, for example, although SSIM appears higher, can the authors indicate whether the improvements are clinically meaningful?

The authors state that their approach favours sampling the low frequencies. What are the consequences of this?

**Detailed Comments:**

- "DL methods excel in reconstructing MRI images from highly-accelerated measurements" -> The measurements are not accelerated. The scan is highly accelerated.

- "The resulted image" -> The resulting image

- "we evaluated each setups" -> we evaluated each setup

**Justification Of Final Rating:**

The authors addressed many of my comments. I disagree that a more detailed validation is out of scope for the work. Though quantitative metrics are important and commonly used in the field, many works have shown that they can be unreliable (Mason et al, IEEE TMI, 2020).

The differences in metrics, especially SSIM, are quite small, and further validation of the reconstruction would have strengthened the paper.

**Justification Of The Preliminary Rating:**

It is a well-written paper with a solid theoretical description and very detailed experiments and comparisons against other methods in the literature. By making the code available, the work is also reproducible.

**Questions To Address In The Rebuttal:**

- Did a medical expert evaluate the reconstructed images?

- Is there a loss of relevant of high frequency information given that the proposed sampling favors the low frequencies?

---

> ### Author Response · Authors · 2026-01-21
> **Responses to Reviewer JQeN**
>
> We thank the reviewer for their careful evaluation of our work and for the constructive feedback. We are encouraged by the positive assessment of the methodological contribution, experimental depth, and reproducibility. Below, we address the raised questions and concerns.
>
> ---
>
> ### 1. Clinical or expert evaluation of reconstructed images
>
> Unfortunately, a clinical evaluation by a radiologist or task-specific downstream assessment was outside the scope of this study and not feasible at the current stage of the project. The primary objective of this work is methodological: to assess whether case-specific, adaptive k-space sampling jointly optimized with reconstruction can improve reconstruction fidelity in accelerated dynamic MRI.
>
> That said, the observed improvements are consistent across quantitative metrics (SSIM, PSNR, NMSE) and statistically significant in the majority of the experimental settings. While we intentionally refrain from making direct clinical claims, such metrics are widely used in the MRI reconstruction literature as proxies for image fidelity and interpretability. For example, results from the CMRxRecon 2024 challenge reported good correlation between radiologist scores and commonly used quantitative reconstruction metrics.
>
> We fully agree that a formal radiological evaluation-particularly in the context of downstream cardiac tasks-would be an important next step, and we view this work as a methodological stepping stone toward such investigations.
>
> ### 2. Low Frequency Prioritization
>
> The learned sampling patterns indeed tend to prioritize low-frequency k-space regions (k-space center), which we interpret as an outcome of the joint optimization rather than an explicit loss design choice. During training, the sampler is supervised indirectly through the reconstruction loss, and sampling low-frequency regions consistently yields larger gains in global image fidelity and temporal consistency under a fixed sampling budget.
>
> At the same time, the model does not exclusively focus on low frequencies. The learned patterns retain coverage of higher spatial frequencies (as illustrated in Figures 4, S8-S13 of the revised manuscript) and this behavior can be linked to the composite loss used during training. While SSIM and L1 primarily encourage accurate recovery of global structure and contrast, the HFEN term explicitly penalizes errors in edge and fine-detail representations, which are commonly associated with high-frequency k-space content, which might justify the occasional sampling of high-frequency components.
>
> ### 3. Code and reproducibility
>
> We agree that public code availability is important for reproducibility and we thank the reviewer for acknowledging that. As stated in the OpenReview submission, the code is available at https://github.com/NKI-AI/direct/tree/adpt .
>
> ### 4. Minor Comments
>
> We thank the reviewer for pointing out the noted typographical and phrasing issues. These have been corrected in the revised version.
>
> ---
>
> We again thank the reviewer for their positive evaluation and insightful questions. We believe that the clarifications above adequately address the concerns raised.

---

> ### Author Response · Authors · 2026-01-29
>
> We thank the reviewer again for the careful evaluation and constructive feedback. We hope that our responses and revisions satisfactorily address the raised questions, and we remain available for any further clarification during the remainder of the discussion period.

---

### Official Review · Reviewer_eTEn · 2026-01-08

**Confidence:** 4
**Preliminary Rating:** 4
**Final Rating:** 5

**Summary:**

The submitted paper titled " End-to-End Adaptive k-space Sampling and Reconstruction for Dynamic MRI" fills astonishing 40 pages, while the main part stays within the 10pages-limit. A an end-to-end deep learning model called  E2E-ADS-Recon is introduced. It jointly optimizes adaptive k-space subsampling and image reconstruction for dynamic MRI - here used for cardiac cine.  The Adaptive Dynamic Sampler (ADS) is presented as a novel approach, which generates case-specific, acceleration-factor-aware sampling patterns. These are optimized either distinct per temporal frame ("frame-specific") or identical across frames ("unified").
The ADS module is trained jointly with generated sensitivity maps (U-net based sensitivity map predictor) and the known reconstruction network vSHARP. The authors use and evaluate their approach on the CMRxRecon multi-coil cardiac cine dataset at acceleration factors of 4×, 6×, and 8× (Cartesian sampling). The main innovation is to evaluate whether learned, case-adaptive sampling patterns improve dynamic MRI reconstruction quality compared to other (fixed, random, or globally optimized) subsampling schemes.

**Strengths:**

The work is very solid. The authors show very deep knowledge of the topic and investigate their approaches thoroughly. Details are fully provided. The core idea of is appealing while in parts a logic next step to known variational k-space sampling schemes, e.g. for accelerated angiography there are products available (https://mriquestions.com/tricks-or-twist.html). This underlines the clinical relevance of the work, in particular to not only improve image quality, but to allow MR-imaging for (completely) new situations and diagnostic questions.
The referenced literature is very comprehensive.

**Weaknesses:**

Overall the amount of experiments and configurations done and tested is too vast in my opinion. Certainly, all the shown results and settings are solid investigations, but in the context of a full conference-paper this not easy to digest, even for experiences and interested readers.
While the adaptive sampling is a newly developed component, the reconstruction module is not.

**Detailed Comments:**

- streamline the paper, remove details
- being an MRI person, I would not call a series of images "imaging sequence" (page 1). In MRI a sequence is usually the "timing sequence for gradients and RF in the acquisition process)
- page 5 dataset: Please specify whether these are all (473) different subjects

**Justification Of Final Rating:**

Thanks the authors for addressing my comments. I have now updated my rating. Although it is not fully in line with other reviewers.
From my position the main topic of the work is worth presenting and valid for future publication.

**Justification Of The Preliminary Rating:**

The work is very solid and detailed. It is relevant, introduces a novel approach and combines it with state-of-the-art methods. All together this is very nice and timely work. Yet, the manuscript is too long and not easy to digest. As it matches the formal page limits, the authors can easily reorganise some sections and further improve to a "strong accept".

**Questions To Address In The Rebuttal:**

The main drafts exactly matched the 10pages limit, however, substantial parts of the results and more importantly a comprehensive discussion are shifted to appendices. In my opinion this is rather suboptimal. The up to 40 pages overall show that many detailed analyses were performed and evaluated, the amount of information exceeds the "acceptance limit" of many readers, I would assume. While I do prefer comprehensive and detailed work, the structure should be optimized to balance methods, results and a broader discussion in the main part. As the performance improvements are partly little, you should also comment on the potential clinical relevance compared to rather consistent (non-adaptive) k-space samplings.
I therefore recommend to restructure parts of the draft, e.g. shift Table 1 (inference times) to the appendix.
The title suggests some novelty also in the reconstruction model, this becomes not apparent to me, maybe remove this from title as the "... Dynamic MRI" anyway suggests that you will reconstruct images as outputs.

---

> ### Author Response · Authors · 2026-01-21
> **Responses to Reviewer eTEn**
>
> We thank the reviewer for the careful reading of our manuscript and for the thoughtful and constructive feedback. We appreciate the positive assessment of the technical depth and relevance of the work. Below, we address the raised points and describe the corresponding revisions.
>
> ---
>
> ### 1. Paper length, structure, and streamlining
>
> We agree that placing the broader discussion exclusively in the appendix was not ideal. Given the post-rebuttal allowance of a 12-page main paper, we have moved the Extended Discussion into the main manuscript. This change allows readers to better contextualize the reported performance gains, understand the trade-offs between adaptive and non-adaptive sampling strategies, and assess practical considerations without having to rely on the supplementary material.
>
> In addition, we have streamlined the presentation of results by moving the inference time analysis (Table 1 in the original submission) to the Supplementary Material, as suggested. We have also slightly streamlined parts of Section 2.3 to reduce low-level technical detail in the main text.
>
> The Supplementary Material remains extensive, but it primarily contains algorithmic descriptions, some implementation details, and mostly additional experiments that are not required to follow the core methodology or conclusions, but may be useful for readers interested in reproducibility or deeper technical inspection. We hope this restructuring improves readability and balances completeness with accessibility.
>
> ---
>
> ### 2. Title and scope of novelty
>
> We would like to clarify that the original title was not intended to suggest architectural novelty in the reconstruction model itself. Throughout the manuscript, including Section 2.4, we explicitly state that the reconstruction backbone is treated as a modular component and instantiated using established state-of-the-art methods (vSHARP, with MEDL-Net used for robustness experiments).
>
> The novelty of the work lies in the end-to-end co-optimization of adaptive, case-specific dynamic k-space sampling and reconstruction, enabled by differentiable sampling and joint training through the end-to-end process. In this sense, “reconstruction” in the title refers to its role within the jointly optimized pipeline rather than to a newly proposed reconstruction architecture.
>
> That said, to avoid potential ambiguity and better align the title with the core contribution, we have slightly revised the title (in the revised manuscript) to:
>
> **“End-to-End Co-Optimization of Adaptive k-space Sampling and Reconstruction for Dynamic MRI.”**
>
> Should we be given the opportunity to update the title in the final version as well, we will adopt this revised wording there too.
>
> We believe this wording preserves the intended scope of the work while making the focus of the contribution more immediately clear.
>
> ---
>
> ### 3. Terminology (“imaging sequence”)
>
> We thank the reviewer for pointing out the terminology issue. We have revised the wording to avoid using “imaging sequence” in a way that could be confused with an MRI pulse sequence, and instead use terminology consistent with MRI conventions when referring to image time series or dynamic frames.
>
> ---
>
> ### 4. Dataset clarification
>
> We now explicitly clarify in the dataset description (Section 3.1) that the 473 scans correspond to distinct subjects.
>
> ---
>
> ### 5. Clinical relevance of observed performance gains
>
> We agree that some of the reported improvements are modest, particularly at lower acceleration factors. This is partly expected given the strength of the reconstruction backbone used, where baseline performance is already high. Nonetheless, the improvements are consistent and statistically significant, and become more pronounced at higher acceleration factors, where non-adaptive sampling schemes degrade more noticeably.
>
> We now discuss this more explicitly in the main paper (Section 5. In addition, we include results at higher acceleration factors (10× and 12×) in Figure S3, obtained by evaluating the trained models beyond the acceleration range used during training. These results follow the same qualitative trends and help illustrate where adaptive sampling provides the clearest benefit.
>
> ---
>
> We thank the reviewer again for the thoughtful feedback. We hope that the clarifications and revisions described above address the raised concerns and improve the clarity of the manuscript.

---

> ### Author Response · Authors · 2026-01-29
>
> We thank the reviewer again for the constructive feedback and for the updated evaluation!

---

### Official Review · Reviewer_baDW · 2026-01-16

**Confidence:** 3
**Preliminary Rating:** 4

**Summary:**

The paper proposes E2E-ADS-Recon, an end-to-end framework that jointly learns scan-adaptive dynamic k-space sampling patterns and a deep reconstruction network for multi-coil cardiac cine MRI. The key component is an Adaptive Dynamic Sampler (ADS) that can produce either frame-specific (varying per time frame) or unified (shared across frames) sampling masks, with support for 1D and 2D Cartesian sampling and multiple acceleration factors trained simultaneously. On the CMRxRecon dataset, the approach consistently outperforms fixed and dataset-optimized non-adaptive baselines, with the largest gains at higher accelerations, and shows similar benefits across two reconstruction backbones (vSHARP and MEDL-Net).

**Strengths:**

1. Introduces a scan-adaptive, dynamic (frame-specific or unified) sampling strategy trained jointly with a state-of-the-art dynamic reconstructor and a sensitivity map predictor; this fills a gap between prior static adaptive sampling and dynamic but non-adaptive learned trajectories.
2. The ADS design is practical and differentiable: encoder+MLP cascades predict per-location probabilities, with budget-rescaling and a straight-through estimator for stochastic binarization, enabling end-to-end training through the sampling step.
3. Supports multi-acceleration training and both 1D (lines) and 2D (points) sampling regimes, systematically comparing frame-specific vs unified strategies.
4. Addresses an important practical limitation in dynamic MRI reconstruction—uniform masks across time—by learning case and frame specific sampling.
5. Demonstrates consistent gains that grow with acceleration, a clinically relevant regime where reconstruction is most challenged.

**Weaknesses:**

1. Evaluation is limited to a single anatomy/dataset (cardiac cine). No test-time generalization to different anatomies or acquisition settings is shown.
2. The STE-based binarization and rejection sampling introduce stochasticity and approximation; sensitivity to the STE surrogate (e.g., slope) and to rescaling choices is not ablated.

**Detailed Comments:**

The end-to-end design is sound: gradients flow from the reconstruction loss to the sampler via a straight-through estimator; budget control via probability rescaling and rejection sampling is standard. Using SENSE reconstructions of progressively acquired data as inputs to the ADS encoder is a reasonable design to condition sampling on already-acquired information.

**Justification Of The Preliminary Rating:**

This paper addresses an important and timely problem: moving beyond uniform, fixed subsampling to scan- and frame-adaptive patterns for dynamic MRI, learned jointly with a strong reconstruction model. The methodology is solid, the evaluation is broad and carefully constructed, and the improvements—especially at higher accelerations—are meaningful. The work advances the state of the art in adaptive sampling for dynamic, multi-coil Cartesian MRI and provides a compelling empirical case that frame-specific adaptivity matters.

The main reservations concern practical feasibility (notably the 2D Cartesian point sampling and lack of hardware constraints), limited evaluation breadth beyond a single dataset/anatomy, and some missing ablations (STE sensitivity, ACS size). Strengthening the discussion of physical implementation and adding a few targeted experiments or clarifications (numerical tables, unseen accelerations, Opt baseline details) would further solidify the contribution. Overall, I find the paper to be of high quality, with clear novelty and value.

**Questions To Address In The Rebuttal:**

1. How sensitive are results to the ACS fraction (e.g., 2%, 6%) and to the STE surrogate (sigmoid slope)? Did you observe stability issues or performance changes with different slopes or deterministic binarization?
2. Could you comment on generalization to other anatomies or motion patterns (e.g., aorta, liver) and whether frame-specific adaptivity brings larger gains when motion is less periodic?

---

> ### Author Response · Authors · 2026-01-21
> **Responses to Reviewer baDW**
>
> Thank you for the careful reading, constructive feedback, and positive assessment of our work. We appreciate the reviewer’s recognition of the novelty and practical relevance of scan- and frame-adaptive sampling for dynamic MRI, and we address the specific questions below.
>
> ---
> ### 1. ACS fraction sensitivity
>
> In all experiments, we fix the ACS fraction to 4% of the sampling space Ω for all methods, including adaptive, learned non-adaptive, and fixed baselines. This choice was made to avoid confounding improvements from adaptive sampling with differences in autocalibration size, and to ensure that all methods operate under identical sensitivity-map estimation conditions. The ACS region is always explicitly acquired and used by the sensitivity map predictor.
>
> In the adaptive setting, sampling decisions begin after ACS initialization (Λ₀ = Λ_ACS unless stated otherwise). Consequently, if additional low-frequency information beyond the ACS were beneficial, the adaptive sampler would need to allocate part of its remaining budget to those locations. Fixing the ACS therefore places the responsibility for subsequent sampling decisions entirely on the adaptive component, rather than encoding this choice manually.
>
> We additionally follow the protocol used by the CMRxRecon challenge [1] where we acquired the data from, which employs a constant ACS fraction across acceleration factors. Finally, we do not expect moderate changes in ACS size (e.g., 2% or 6%) to qualitatively alter the conclusions, as the sensitivity estimation network is trained end-to-end together with the reconstruction model (and ADS where applicable) under identical conditions for all methods.
>
> ---
> ### 2. STE-based binarization and hyperparameter sensitivity
>
> We acknowledge that the straight-through estimator (STE) was not exhaustively ablated. Our implementation follows practice drawn from prior optmized sampling work, where stochastic binarization is used in the forward pass and the backward pass is approximated with a sigmoid surrogate. The slope was fixed to values commonly used in the literature (10 in our implementation, comparable to values used in related work – 12 in [2] and 10 in [3]).
>
> In our experiments, we did not observe instability during training attributable to the STE, and convergence behavior was consistent across runs. That said, we acknowledge that a sensitivity analysis with respect to the surrogate slope would be informative.
>
> ---
> ### 3. Clarification of the “Opt” baseline
>
> By “Opt”, we refer to a dataset-optimized, non-adaptive learned sampling scheme: a single sampling mask is learned end-to-end on the training set and then applied uniformly at test time. This corresponds to prior work such as LOUPE [3, 4], which learns a global sampling pattern jointly with a reconstruction network.
>
> In our experiments, we evaluate this baseline within the same overall pipeline as all other methods. The reconstruction backbone, sensitivity map predictor, and training settings are identical; only the sampling parameterization differs. We also extend this baseline to frame-specific or unified sampling settings for fair comparison, while retaining its non-adaptive nature at test time.
>
> ---
>
> ### 4. Generalization to different anatomies
>
> While the main evaluation focuses on multi-coil cardiac cine MRI, we agree that generalization beyond a single anatomy is important. To partially address this, we include additional results on an unseen ventricular outflow tract / aortic cine dataset from CMRxRecon 2025 [5], which exhibits a different anatomical view and motion profile and was not observed during training.
>
> The observed trends are consistent with those reported in the main experiments, supporting the broader applicability of adaptive sampling in dynamic settings. These results have been added to the revised manuscript (Figure S4).
>
> ---
> ### 5. Generalization to unseen accelerations
>
> Models were trained using acceleration factors drawn from {4×, 6×, 8×}. To address the reviewer’s comment we have now additionally evaluated performance at higher accelerations than used during training (10× and 12× unseen during training) to examine extrapolation behavior. The results follow the same qualitative pattern observed at lower accelerations. These results are included in the revised manuscript (Figure S3).

---

> ### Author Response · Authors · 2026-01-21
> **Responses to Reviewer baDW (continued)**
>
> ### 6. Hardware Constraints and 2D sampling
>
> Our study is retrospective and focuses on demonstrating the potential of end-to-end co-optimization of sampling and reconstruction for dynamic MRI acquisitions. We agree that prospective implementation on MRI hardware, particularly for 2D Cartesian point sampling, introduces practical constraints while they remain outside the scope of this work.
>
> Nevertheless, the retrospective results provide insight into what could be gained from adaptive strategies if such constraints were addressed. Consistent with prior work, we observe that 2D sampling outperforms 1D sampling in this setting too, and we discuss the implications and limitations of this observation in the revised discussion without making claims about immediate clinical deployability.
>
> ---
>
> We have revised the manuscript accordingly. We thank the reviewer again for their thoughtful and constructive comments and hope that these clarifications adequately address the raised questions and concerns.
>
> ---
>
> ### References
>
> 1. Lyu, Jun, et al. "The state-of-the-art in cardiac mri reconstruction: Results of the cmrxrecon challenge in miccai 2023." Medical Image Analysis 101 (2025): 103485.
> 2. Zhang, Jinwei, et al. "Extending LOUPE for K-space Under-sampling Pattern Optimization in Multi-coil MRI." International Workshop on Machine Learning for Medical Image Reconstruction. Cham: Springer International Publishing, 2020.
> 3. Bakker, Tim, et al. "On learning adaptive acquisition policies for undersampled multi-coil MRI reconstruction." International Conference on Medical Imaging with Deep Learning. PMLR, 2022.
> 4. Bahadir, Cagla Deniz, Adrian V. Dalca, and Mert R. Sabuncu. "Learning-based optimization of the under-sampling pattern in MRI." international conference on information processing in medical imaging. Cham: Springer International Publishing, 2019.
> 5. Daryl Xu, "CMRxRecon2025", IEEE Dataport, February 25, 2025, doi:10.21227/b6xs-gv29

---

> ### Author Response · Authors · 2026-01-29
>
> We thank the reviewer again for the careful and constructive feedback. We hope that our responses and revisions address the raised points satisfactorily, and we remain available for any further clarification during the remainder of the discussion period.

---

### Author Rebuttal · Authors · 2026-01-21

**Rebuttal:**

We thank all reviewers for their careful assessment of our manuscript and for the positive and constructive feedback. We appreciate the time invested in reviewing our work and the thoughtful suggestions provided.

We have addressed the reviewers’ comments where relevant and provide detailed responses to each point.

For the rebuttal, we have uploaded a single ZIP file containing the following materials:

•	**Revised Manuscript.pdf** -the updated version of the paper

•	**Revised Manuscript with Highlighted Changes.pdf** -the revised manuscript with all modifications clearly marked

•	**Responses to Reviewers.pdf** -a detailed, point-by-point response addressing each reviewer’s comments individually.


In addition, we have responded directly to each reviewer’s comments using the Official Comment feature in the review system.

Thank you again for the positive evaluation and constructive feedback and we hope you find our responses satisfactory.

**Supporting Material:**

/attachment/ffedc8ee94d5b33a97da0383a378a811a0e1181b.zip

---

### Meta-Review · Area_Chair_CFg7 · 2026-02-04

**Recommendation:** Accept (Poster)
**Confidence:** 5

**Metareview:**

All reviewers found the proposed method to be novel and the results promising.

---

### Decision · Program_Chairs · 2026-02-13

Accept (Poster)